



# First release of the Pelagic Size Structure database: Global datasets of marine size spectra obtained from plankton imaging devices

Mathilde Dugenne [1,*], Marco Corrales-Ugalde [2*], Jessica Y. Luo [3], Rainer Kiko [1,4], Todd D. O'Brien [5], Jean-Olivier Irisson [1], Fabien Lombard [1], Lars Stemmann [1], Charles Stock [3], Clarissa R. Anderson [6], Marcel Babin [7], Nagib Bhairy [8], Sophie Bonnet [8], Francois Carlotti [8], Astrid Cornils [9], E. Taylor Crockford [10], Patrick Daniel [11], Corinne Desnos [1], Laetitia Drago [1], Amanda Elineau [1], Alexis Fischer [11], Nina Grandrémy [12], Pierre-Luc Grondin [7], Lionel Guidi [1], Cecile Guieu [1], Helena Hauss [4, 13], Kendra Hayashi [11], Jenny A. Huggett [14,15], Laetitia Jalabert [1], Lee Karp-Boss [16], Kasia M. Kenitz [5], Raphael M. Kudela [11], Magali Lescot [8], Claudie Marec [7], Andrew McDonnell [17], Zoe Mériguet [1], Barbara Niehoff [9], Margaux Noyon [18], Thelma Panaïotis [1], Emily Peacock [10], Marc Picheral [1], Emilie Riquier [1], Collin Roesler [10], Jean-Baptiste Romagnan [12], Heidi M. Sosik [10], Gretchen Spencer [7], Jan Taucher [4], Chloé Tilliette [1], and Marion Vilain [1,19]

[1]Sorbonne Université, CNRS, Laboratoire d'Océanographie de Villefanche, Villefranche-sur-mer, France
[2]Atmospheric and Oceanic Sciences, Princeton University, Princeton, NJ, USA
[3]NOAA/OAR Geophysical Fluid Dynamics Laboratory, Princeton, NJ, USA
[4]GEOMAR Helmholtz Centre for Ocean Research Kiel, Kiel, Germany
[5]NOAA Fisheries - Office of Science and Technology, Silver Spring, Maryland, USA
[6]Southern California Coastal Ocean Observing System, Scripps Institution of Oceanography, University of California San Diego, La Jolla, California
[7]Takuvik International Research Laboratory, Quebec Ocean, Laval University (Canada) - CNRS, Departement de biologie and Quebec-Ocean, Universite Laval, Quebec, Canada
[8]Mediterranean Institute of Oceanography
[9]Polar Biological Oceanography, Alfred Wegener Institute Helmholtz Centre for Polar and Marine Research, Bremerhaven, Germany
[10]Biology Department, Woods Hole Oceanographic Institution, Woods Hole, MA, USA
[11]Ocean Sciences Department, University of California, Santa Cruz, Santa Cruz, CA, USA
[12]DECOD (Ecosystem Dynamics and Sustainability), IFREMER, INRAE, Institut Agro, Nantes, Centre Atlantique - Rue de l'Ile d'Yeu - BP 21105, 44311 Nantes Cedex 03, France
[13]NORCE Norwegian Research Centre, Norway
[14]Oceans and Coastal Research, Department of Forestry, Fisheries and the Environment, Cape Town, South Africa
[15]Department of Biological Sciences, University of Cape Town, South Africa
[16]University of Maine, USA
[17]Oceanography Department, University of Alaska Fairbanks, Fairbanks, AK, United States
[18]Department of Oceanography and Institute for Coastal and Marine Research, Nelson Mandela University, Gqeberha, 6001, South Africa
[19]Current address: Laboratoire de Biologie des Organismes et Ecosystèmes Aquatiques, Muséum National d'Histoire Naturelle, CNRS, IRD, SU, UCN, UA, Paris, France

**Correspondence:** Mathilde Dugenne (mathilde.dugenne@imev-mer.fr), Marco Corrales-Ugalde (mcugalde88@gmail.com)

**Abstract.** In marine ecosystems, most physiological, ecological, or physical processes are size-dependent. These include metabolic rates, uptake of carbon and other nutrients, swimming and sinking velocities, and trophic interactions, which



eventually determine the stocks of commercial species, as well as biogeochemical cycles and carbon sequestration. As such, broad scale observations of plankton size distribution are important indicators of the general functioning and state of pelagic
ecosystems under anthropogenic pressures. Here, we present the first global datasets of the Pelagic Size Structure database (PSSdb), generated from plankton imaging devices. This release includes the bulk particle Normalized Biovolume Size Spectrum (NBSS) and bulk Particle Size Distribution (PSD), along with their related parameters (slope, intercept, and $R^2$) measured within the epipelagic layer (0-200 m) by three imaging sensors: the Imaging FlowCytobot (IFCB), the Underwater Vision Profiler (UVP) and benchtop scanners. Collectively, these instruments effectively image organisms and detrital material in the
7-10,000 $\mu$m size range. A total of 92,472 IFCB samples, 3,068 UVP profiles, and 2,411 scans passed our quality control and were standardized to produce consistent instrument-specific size spectra averaged in 1x1° latitude/longitude, and by year and month. Our instrument-specific datasets span all major ocean basins, except for the IFCB which was exclusively deployed in northern latitudes, and cover decadal time periods (2013-2022 for IFCB, 2008-2021 for UVP, and 1996-2022 for scanners), allowing for a further assessment of the pelagic size spectrum in space and time. The datasets that constitute PSSdb's first
release are available at https://doi.org/10.5281/zenodo.10150020 (Dugenne et al., 2023).

# 1 Introduction

## 1.1 The relevance of plankton size to approximate ecological processes

Plankton size-structure observations are essential to bridge the gap between marine biogeochemical processes and biological stock assessments, including those of important commercial species (Boyd and Newton, 1999; Armstrong et al., 2001;
Finkel et al., 2009; Guidi et al., 2009; Taniguchi et al., 2014; Hillebrand et al., 2022). Historically, ecosystems dominated by small phytoplankton were thought to support regenerated production, rapidly recycled in the epipelagic layer, and to contribute little to carbon export. Conversely, larger phytoplankton were thought to fuel higher trophic levels and contribute to a large extent to carbon sequestration by sinking relatively fast to the mesopelagic (200-1000 m) layers (Legendre and Le Fèvre, 1995; Wassmann, 1997; Durkin et al., 2015). Although this paradigm has shaped almost all current biogeochemical models and their
projections of marine ecosystem services under climate change, recent studies have challenged this concept. Indeed, plankton of intermediate size and/or trophic levels have been shown increasingly to contribute significantly to biogeochemical functioning and carbon export (Lomas and Moran, 2010; Choi et al., 2014; Durkin et al., 2015; Guidi et al., 2016; Ward and Follows, 2016; Biard et al., 2016; Leblanc et al., 2018; Richardson, 2019; Juranek et al., 2020; Schvarcz et al., 2022). These studies call for a global assessment of the plankton size continuum, rather than discrete size categories, to study ecosystem functioning or
to model ecosystem services under current and future anthropogenic pressures (Lombard et al., 2019; Ljungström et al., 2020; Atkinson et al., 2021).

The first estimates of plankton and particle size spectra across several orders of magnitude yielded global and robust patterns of roughly equal amounts of biomass distributed across particle sizes (Sheldon et al., 1972). Since this seminal study, there has been increasing recognition that plankton size structure is an effective way to summarize the inherent complexity of
community structure (Stemmann and Boss, 2012) and how it relates to key ecosystem processes such as primary productivity



(Marañón et al., 2001), fishery yields (Sheldon et al., 1977) and sequestration of carbon dioxide ($CO_2$) from the atmosphere (Basu and Mackey, 2018). This is possible because organism body size serves as a "master trait" from which other biological properties are derived, such as metabolism (Huete-Ortega et al., 2012; Ikeda, 2014; Kiørboe and Hirst, 2014; Maas et al., 2021), growth rates (Hopcroft et al., 1998; Chen and Liu, 2010; Edwards et al., 2012), consumption rates (Hansen et al., 1994;

Kiørboe and Hirst, 2014), predator-prey size ratios (Hansen et al., 1994; Hauss et al., 2023), mortality (Hirst and Kiørboe, 2002), active transport through diel vertical migration (Ohman and Romagnan, 2016), and sinking (Smayda, 1971; Cael et al., 2021). These size-dependent processes have been historically represented by allometric relationships, also referred to as power law functions, whose parameters were derived empirically (see review from Chisholm (1992) and Hillebrand et al. (2022)) or mechanistically (see review from Andersen et al. (2016)). Given the use of plankton and particle size structure as a proxy for

complex ecological processes, estimates of pelagic size structure, with large spatial and temporal coverage, are essential to assess ecological trends across space and time.

## 1.2 The emergence of marine imaging devices and size structure observations

The need to capture pelagic size spectra at unprecedented scales has sparked the emergence of a multitude of *in situ* and laboratory-based plankton imaging systems in the past 20 years, with individual instruments designed to capture the

continuous size distribution of organisms and detrital particles in a specific size range (Davis et al. (2005); Olson and Sosik (2007); Gorsky et al. (2010); Picheral et al. (2010); Sieracki et al. (2010); Ohman et al. (2019)). Plankton large enough to be identified and sized at the resolution of commercially available imaging systems include (1) nano- and microplanktonic protists (comprising photoautotrophs, mixotrophs, and heterotrophs), typically imaged by the FlowCam (Sieracki et al., 1998) or the IFCB flow cytometer (Sosik and Olson, 2007), (2) micro-, meso- and macroplankton (comprising large chain-forming

photoautotrophs, mixotrophs, and heterotrophs), routinely imaged *in situ* by UVPs (Picheral et al., 2010; Stemmann et al., 2012), CPICSs (Gallager, 2016), or VPRs (Davis et al., 2005), or collected with nets and later imaged onboard with a ZooCAM (Colas et al., 2018), or in the lab with benchtop scanners like the ZooScan (Gorsky et al., 2010; Lehette and Hernández-León, 2009; Kiko et al., 2020), as well as (3) micronekton, which can complement the size range of meso- and macroplankton, well detected by ISIIS instruments (Cowen and Guigand, 2008). Collectively, these imaging systems can capture a wide size range

of marine plankton, spanning a few micrometers to tens of centimeters (Lombard et al., 2019), providing accurate estimates of plankton community structure and trophic dynamics (Atkinson et al., 2021). More recently, they also provided insight into diverse detrital pools, which comprise fecal pellets, deadfalls, or marine snow aggregates linked to specific biogeochemical properties (Kiko et al., 2017; Trudnowska et al., 2021). Such particles generally dominate UVP images across all size classes (Stemmann and Boss, 2012; Kiko et al., 2022), highlighting yet another continuum in particle transformation and degradation

(Durkin et al., 2021). As part of the digital revolution, these advancements in new technologies have been matched with an equally rapid diversification in sampling strategies (e.g., towed-, net-, moored-, or profiling-based sampling), available platforms (e.g., floats, gliders, buoys, moorings, ships of opportunity, research vessels), data processing and management tools (e.g., collaborative platforms for image classification like EcoTaxa), or automated taxonomic (Luo et al., 2018; Irisson et al.,





2022) and functional (Schröder et al., 2020; Orenstein et al., 2022) classification schemes, such that plankton imaging systems
have become widespread for research and monitoring applications alike.

Phytoplankton and zooplankton biomass and diversity, as well as bulk particulate matter, were identified as essential ocean, biodiversity, and climate variables by the Global Observing Systems (Chiba et al., 2018; Batten et al., 2019). Thus, plans are now underway to measure these variables on large-scale observing programs, like the Bio-GO-SHIP (Clayton et al., 2022) or the BGC-Argo (Claustre et al., 2020; Picheral et al., 2022) programs, using the IFCB and the UVP6 (Picheral et al., 2022), respec-

tively. Other observing programs include long-term time-series using IFCB (e.g. California Ocean Observing System (CalOOS, https://data.caloos.org/) and the Northeast U.S. Shelf Long-Term Ecological Research (NESLTER, https://nes-lter.whoi.edu/)) or ZooScan (e.g. Point B in the Bay of Villefranche), which are constrained spatially but can resolve temporal trends with great accuracy. More recently, the combination of ZooScan and ZooCAM (Grandrémy et al., in press) has enabled the analysis of a regional scale, long-term zooplankton survey (2004-2019, ongoing) of a temperate European continental shelf (Grandrémy

et al., 2023a, b, c). Overall, sustained observations from IFCBs and UVPs have been ongoing since 2006/8 respectively, and even track back to 1966 for laboratory-based ZooScan observations from preserved samples (García-Comas et al., 2011). Despite such time spans, Kiko et al. (2022) only recently published the first curated global dataset of particle size between 64-50,000 $\mu$m, obtained from UVP5 measurements solely. This release was facilitated by a collaborative management platform, EcoPart (https://ecopart.obs-vlfr.fr), which enables the collection of count and size information of bulk particles detected by the

UVP. This unique platform, along with other collaborative platforms such as EcoTaxa (https://ecotaxa.obs-vlfr.fr), the IFCB dashboards (https://ifcb-data.whoi.edu/dashboard, https://ifcb.caloos.org/dashboard) and their corresponding application programming interfaces (API) allowed to find and access size structure estimates easily and repeatedly, two of the FAIR (Findable, Accessible, Interoperable, Reusable) data principles guiding current data management strategies (Lombard et al., 2019).

### 1.3  The Pelagic Size Structure database project

With the support of many international data providers, we developed the Pelagic Size Structure database (PSSdb, https://PSSdb.net) in accordance to FAIR principles (Wilkinson et al., 2016) to provide global datasets of particle and plankton size distributions. Our project capitalizes on largely untapped size structure observations from plankton imaging devices, consistent across the 7-10,000 $\mu$m size range (Romagnan et al., 2015; Lombard et al., 2019), and aims to become a global data source like the NOAA World Ocean Database (https://www.ncei.noaa.gov/products/world-ocean-database) and COPEPOD

(https://www.st.nmfs.noaa.gov/copepod). The objectives for PSSdb were both to (1) implement a workflow able to retrieve counts, sizes, and taxonomic information from online imaging data streams to calculate particle size spectra, and to (2) provide multi-level, harmonized products, matching the spatio-temporal resolution of current biogeochemical models. Our workflow is programmed in Python and can be fully tuned to specific instruments, spatio-temporal resolutions and research questions, such as mesoscale plankton distribution, patchiness, short-term trophic dynamics or diel vertical migration, with little modification.

To achieve this, we favoured a general framework to estimate size spectra from existing data sources, that can also be updated with new data from current and new technologies. Expected products will range from low (bulk particles and planktonic size spectra, presented in this paper) to high taxonomic resolution matching the functional groups in biogeochemical models.





Currently, our pipeline include size spectra estimates from two widespread, synoptic approaches, the Particle Size Distribution (PSD) and the Normalized Biovolume Size Spectrum (NBSS), developed by ecologists and optics scientists in the mid 1960s and 1990s to summarize and link size structure to ecosystem properties, communities, and ecological processes (Sheldon et al., 1972; Jonasz and Fournier, 1996; Kostadinov et al., 2009; Stemmann et al., 2012; Sprules and Barth, 2016). Both metrics have been adopted to represent the exponential decrease in particle abundance as size increases, with abundance traditionally expressed as either normalized particle number or biovolume/biomass, respectively. This exponential decrease in abundance with size is mostly linear when transformed to a logarithm scale (Sheldon et al., 1977), unless abiotic or biotic perturbations lead to local peaks of intermediate-size organisms (Moscoso et al., 2022). Both the slope and intercept of the log-linear regression between particle abundance and size are important indicators of pelagic ecosystem changes(Sprules and Munawar, 1986). They represent the equilibrium between lower and upper trophic levels, which can be indicative of trophic transfer efficiency, and the ecosystem carrying capacity, respectively (Zhou, 2006). In this paper, we present the first version of PSSdb instrument-specific datasets, consisting of bulk size spectra and derived parameters (slope, intercept and $R^2$) measured by the IFCB, the UVP and benchtop scanners (e.g. ZooScan) within the epipelagic layer. First, we highlight the large spatio-temporal coverage of our observations, before describing the shape of the size spectra and patterns of their derived parameters. Finally, we discuss how PSSdb provides a way to study the links between plankton community structure and global biogeochemical fluxes, and thus inform the development of biogeochemical and data-driven models.

## 2 Materials and methods

In the following sections, we first describe the data acquisition and processing steps prior to PSSdb ingestion for the three imaging approaches used for this release (section 2.1). Then we provide details on the current pipeline for PSSdb ingestion that enables the computation of instrument-specific size spectra currently available at https://doi.org/10.5281/zenodo.10150020 (section 2.2).

### 2.1 Acquisition and pre-processing of IFCB, UVP, and scanner imaging datasets

#### 2.1.1 Imaging FlowCytobot (IFCB)

The IFCB is a submersible flow cytometer coupled to a microscope camera, with an effective resolution of either ~2.77 or ~3.44 pixels per $\mu$m, depending on the segmentation threshold used to extract morphometric measurements. According to the camera resolution, IFCB instruments may detect particles in the 4-420 $\mu$m size range (Olson and Sosik, 2007). In continuous mode, individual samples with a 5 mL maximum volume are automatically drawn by a syringe approximately every 20 min. Instruments can be deployed on underwater moorings (down to 40 m depth), on land-based piers and wharves, or on research vessels, where they can be connected to the flow-through system of the vessel to automatically collect new samples throughout the cruise. Alternatively, they may also be used to analyse discrete samples obtained from Niskin bottles from the CTD-Rosette, though in general, most IFCB sampling efforts are limited to a single depth, located within the mixed layer (Suppl. A1). In



this instrument, a sheath fluid is recycled continuously through a set of two cartridge filters to align single, colonial, or chain-
forming particles and drive them through the flow cell, where they are intercepted by a red laser beam (630 nm). The resulting
scattering and fluorescence emissions are captured and transformed by photo-multipliers (PMT), whose function is to amplify
(depending on the PMT relative gain set) and convert the emitted photons into an electronic signal. Image acquisition may
be triggered by either scattering or fluorescence, given the individual gain and threshold set by the instrument user prior to
sampling, if the particle size exceeds a minimum area threshold (>160 pixels or ∼4 $\mu$m in equivalent circular diameter). Raw
IFCB data include the individual images detected in real-time (.roi files), the summary statistics of the electronic PMT signals
(.adc files), and the configuration settings (.hdr files). The morphometric measurements, including image area, feret diameter,
and biovolume estimates based on distance map matrices (Moberg and Sosik, 2012), of individual or multiple (in the case
of chain-forming or colonial organisms) Region Of Interests (ROI) are extracted from the masked images (also referred to
as blobs) using custom feature extraction Matlab code (code and documentation available at: https://github.com/hsosik/IFCB-
analysis/) and can be further used to predict taxonomic annotations (Sosik and Olson, 2007). Taxonomic annotations were used
to remove artifacts before data ingestion into PSSdb, and will allow for further work on taxon-specific data products for future
releases .

### 2.1.2 Underwater Vision Profiler (UVP)

The 5th generation of UVP (hereafter, UVP5) consists of a system of two red LED lights (625 nm) that illuminates a
22x18 cm frame, which is imaged by a ∼8 pixels per mm resolution camera facing the illuminated plane. This 6000 m depth-
rated system has been routinely mounted on CTD-Rosettes (Picheral et al., 2010), before its miniaturization led to the next
generation of UVPs (UVP6, Picheral et al. (2022)), also 6000 m depth-rated. UVP6 instruments only have one red LED light
and image a smaller frame (15x18 cm) with a higher resolution (∼12 pixels per mm). As a result of its miniaturization, the
UVP6 can be mounted on autonomous platforms like gliders, floats, or moorings to record images at a preset time interval,
although acquisitions have mostly been done in profiling mode so far (Suppl. A1). On the descent, pressure sensor readings
and images are recorded at a frequency of 6 to 20 Hz, depending on the configuration setting and the *in situ* concentration
of particles, whereby low concentrations require less buffering time before each new acquisition and hence allow a higher
acquisition frequency. The configuration setting allows users to record the raw image frames, the vignettes of particles larger
than a fixed size threshold generated after segmentation (i.e. the process of extracting individual ROIs from the initial image),
or a combination of both (full process mode). The size threshold is typically set to 44±22 pixels (∼910±80 $\mu$m in equivalent
circular diameter, or ECD) and 70±15 pixels (∼690±120 $\mu$m in ECD) for the UVP5 and UVP6, respectively. In mixed
acquisition mode (the recommended setting to limit processing time during and post-deployment), image frames are segmented
in real-time to extract individual area and mean gray level estimates for each particle larger than 1 pixel (∼150±30 and ∼80±10
$\mu$m in ECD for UVP5 and UVP6, respectively) and vignettes of larger particles are saved as bmp thumbnails. Post-recovery,
the metadata are manually filled and the vignettes' bmp files are converted to binary masks whose morphometric features,
including area and ellipsoidal axis, are extracted by a custom ImageJ toolbox named Zooprocess (Gorsky et al., 2010) for the
UVP5 or via the UVPapp for the UVP6 (Picheral et al., 2022). Size estimates for all particles can be further stored in EcoPart





(https://ecopart.obs-vlfr.fr), while vignettes can be uploaded to the collaborative platform EcoTaxa (https://ecotaxa.obs-vlfr.fr), for automatic class predictions and manual validation. Prior to instrument shipping, both the effective volume (0.98±0.18 L

for UVP5 and 0.6±0.02 L for UVP6) of the image frame and the two size conversion factors, Aa (the intercept) and Exp (the slope), linking metric-based to pixel-based area estimates by a power-law function, are calibrated against the unique reference unit (Picheral et al., 2010, 2022). However, the size conversion factors are used to account for light scattering around small particles only, but are not required for size estimates of large particles, and the use of these factors can result in larger error propagation compared to a fixed pixel size conversion factor (data not shown). Therefore, all pixel-based area estimates were

converted to metric area using a fixed pixel size factor (corresponding to the camera resolution reported above) for the UVP data included in the current PSSdb version. For further details regarding UVP data processing see Kiko et al. (2022).

### 2.1.3 Net-sampling and benchtop scanners

Traditionally, zooplankton samples are collected via a wide range of net systems (reviewed by Wiebe and Benfield (2003)), preserved with a fixative reagent (mostly a buffered formaldehyde seawater solution) and processed in the laboratory. Benchtop

flatbed scanning systems allow for a relatively high sample throughput compared to the traditional microscopic approach. PSSdb currently includes data collected from vertical or oblique tows with nets of various mesh sizes and aperture diameters (Suppl. A1), mostly equipped with flow-meters, and analysed with the ZooScan system (Gorsky et al., 2010) or alternative generic scanner (Gislason and Silva, 2009; Lehette and Hernández-León, 2009; Kiko et al., 2020). These benchtop scanners have a resolution of ∼96 pixels per mm respectively, with the frame illuminated from above and scanned from below. Both

are typically used to scan and digitize preserved zooplankton samples, as the organisms must be immobile during scanning. Prior to scanning, a background image of the frame filled with distilled water is scanned to facilitate ROI segmentation. The samples are typically rinsed to remove the fixative and the seawater, size-fractionated using sieves of various mesh sizes, and subsampled into aliquots to reduce the number of organisms per scan and to avoid overlapping objects in the image (Jalabert et al., 2022). Similarly to UVP5 profiles, Zooprocess is used to save the scanner frame and manually fill the metadata of each

sample, including the GPS coordinates, the sampling depth range, sampling time, volume of filtered seawater and the dilution factor of the scanned subsamples. Each scan will generate three files, containing the log, metadata and the overall scan saved as tiff files. A first segmentation is performed to separate the ROIs from the background, and extract their morphometric features (see suppl. material of Gorsky et al. (2010)), depending on a lower size threshold (370±360 $\mu$m in ECD on average) and the mean gray level intensity (default is 243). If necessary, a second segmentation may be done after manually separating

overlapping ROIs (Vandromme et al., 2012). Once the separation of ROIs is optimal, their corresponding vignettes, along with the automatically generated EcoTaxa table, may be uploaded to EcoTaxa to predict and validate the taxonomic annotations. As a starting point, and for reproducibility, we only ingested datasets uploaded on EcoTaxa, as they can be repeatedly accessed and shared amongst collaborators, notably to assess the annotation status, which is important for ingestion into PSSdb (see section 2.2.4). Once datasets are exported from EcoTaxa, we consider the reported size-based fractionation of the net tow sample:

if the sample was sieved into separate size fractions after the collection, (i.e. a sample collected with 333$\mu$m mesh net that was afterwards sieved through 150$\mu$m, 500$\mu$m and 1 mm meshes) the size spectra are first calculated for each size fractions



based on the dilution factor of the aliquots taken for each sieved sample "acq_sub_part" column in EcoTaxa) and the volume of filtered sea water of the net (as determined by the flowmeter; "sample_tot_vol" column in EcoTaxa), to account for the volume effectively scanned within a size fraction. The total size spectrum is then obtained by summing the fraction-specific spectra, since size fractionated scans originate from the same volume.

## 2.2 PSSdb data pipeline

The current PSSdb pipeline is illustrated in Figure 1 and includes 5 major steps: the selection (section 2.2.1) and extraction (section 2.2.2) of imaging datasets from online data streams, the standardization (section 2.2.3) and quality control and of the datasets (section 2.2.4), the binning of instrument-specific datafiles (section 2.2.5), and lastly, the computation of particle size spectra and derived parameters (section 2.2.6).

### 2.2.1 Selection of imaging data streams

The first objective of the Pelagic Size Structure database (PSSdb) is to estimate particle size spectra from plankton imaging devices, following the FAIR principles. We thus relied primarily on online and accessible platforms created by the instrument developers to manage their datasets: IFCB dashboards (of generation 2 exclusively, as generation 1 does not include metadata like longitude and latitude) and EcoTaxa/EcoPart, developed for ZooScan and UVP, but also for IFCB and other imaging systems, since a few IFCB datasets ingested in PSSdb were available on EcoTaxa. IFCB dashboards are deployed by individual labs or regional networks with specific urls and are publicly accessible. Conversely, EcoTaxa datasets are not accessible by default, so data providers who wanted to contribute to the PSSdb were asked to provide access to their projects.

Both IFCB dashboards and EcoTaxa contain sample metadata, raw images, their related morphometric measurements and optionally, their taxonomic annotation. To ensure that size distributions were representative of living and non-living particles and excluded methodological artefacts, we selected datasets with predicted and/or curated image classification. Of the 37 datasets on the IFCB dashboards and the 3,290 UVP, scanner, and IFCB datasets on EcoTaxa (last checked on Oct 2023), only 6 projects from IFCB dashboards and 250 projects from EcoTaxa were downloaded and integrated into the first PSSdb release products. The list of the datasets (and their urls) that are included in the first PSSdb release can be downloaded at https://doi.org/10.5281/zenodo.10150020, in the "data sources" spreadsheet included in the compressed release datafiles. The dataset list was generated automatically using the EcoTaxa and IFCB dashboards Application Programming Interface (API), which also provides fast and automatic access to both data (morphometric measurements and taxonomic annotation) and metadata.





**Figure 1.** Schematic of the PSSdb processing pipeline. The main steps of the pipeline include (a) the selection and automatic download of imaging datasets, (b) the standardization of their native formats and units, (c) a quality control involving an exchange between PSSdb developers and the concerned principal investigators, (d) the binning of samples in spatio-temporal proximity to match the current resolution of other databases and biogeochemical models, and (e) the computation of size spectra and generation of the data products released at https://doi.org/10.5281/zenodo.10150020.



### 2.2.2 Extraction of imaging data streams

All functions to list (funcs_list_projects.py) and export (funcs_export_projects.py) datasets from IFCB dashboards and EcoTaxa/EcoPart automatically are available at https://github.com/jessluo/PSSdb/tree/main/scripts. To export IFCB datasets, sample-specific queries to the IFCB dashboards are executed sequentially to retrieve sample metadata such as location, time, and depth, plus the morphometric measurements of individual ROIs stored in the "features" files, and the top 5 taxonomic predictions, stored in "autoclass" files. Metadata, feature, and autoclass files are then combined in a single master table, with a

row for each ROI, and saved into multiple files comprising ∼ 500,000 rows to limit the size of the exports and the processing time.

Scanner and UVP datasets were automatically exported from EcoTaxa using the API with the default option. This option retrieves all the information relative to individual ROIs (e.g. area, taxonomic annotation) and samples (e.g. location, depth, time), as well as specific acquisition (e.g. size fraction scanned and associated volume), and processing (e.g. pixel calibration

factor) steps.

To further retrieve the size and count information of small particles processed by UVPs in real time, which are only uploaded to EcoPart, we wrote a custom script based on existing web scraping python modules. We selected the "raw" export option for all datasets hosted on EcoPart, rather than the default export option which provides summary statistics, consisting of the summed particle counts and biovolume in individual size bins, computed in 5 m depth bins. With the raw export option, we

were able to retrieve the number of particles (column "nbr") of a given pixel-based size measurement (column "area"), as well as the number of individual image frames (column "imgcount", used to calculate the cumulative volume) in 1 m depth bins. This strategy has multiple advantages, as it allows the conversion of pixels to metric area estimates using either the power-law function described in section 2.1.2 or a fixed pixel size. It also allows for the construction of size spectra using custom size bins, and for an assessment of the uncertainty of the size spectra estimates, using the bootstrap approach published by Schartau

et al. (2010).

Using a pair of identifiers allowing to link each UVP dataset uploaded on EcoPart to its corresponding EcoTaxa ID, the datasets on both platforms were consolidated into a single table to account for all particles detected by the UVP. Since EcoPart "raw" datafiles are summarized in 1 m depth bins, it is impossible to link a specific area estimate to the corresponding EcoTaxa vignette, and thus, its taxonomic annotation. To consolidate data for all particles in 1 m depth bins without losing further

information and without including the same particle twice, we used the threshold for vignette generation to select particles with and without a taxonomic annotation (particles larger than ∼910 or 690 $\mu$m in ECD for the UVP5 and UVP6, respectively). The consolidated UVP datafiles thus include the area estimates from all particles smaller than this threshold, which are assigned an empty taxonomic annotation, along with the area and taxonomic annotations of each ROI stored in EcoTaxa datafiles, whose sampling depth precision is reduced to the resolution of EcoPart datafiles (i.e. 1 m bin levels). All metadata for the sampling

locations, depth range, and pixel size were merged to this unique table using the metadata file exported from EcoPart.



### 2.2.3 Standardization of imaging datasets

Since raw datasets exported from the API queries are generated with different formats, with specific headers and units, we developed instrument-specific "standardizer" spreadsheets in order to re-format all datasets to the same standard. Each spreadsheet contains the dataset IDs for a given instrument, including the pair of IDs required to consolidate UVP datasets (see

section 2.2.2), and the information required for the standardization and quality control of these datasets. The dataset ID lists are generated automatically, but the data information (headers and units) are manually filled to map the native headers and units of the datafiles to standard names (following the variablename_field nomenclature) and units (following the variablename_unit nomenclature). These spreadsheets can be downloaded at https://github.com/jessluo/PSSdb/tree/main/raw.

The mapped variables include longitude, latitude, sampling time (with time format), minimum and maximum sampling

depth, volume sampled and potential dilution factors, the lower and upper sample size limit, and optional additional metadata describing the sampling effort, protocol or downstream processing, the pixel size, and the ROI size estimates with taxonomic annotation. In the case of size-fractionated samples, the sampling size limits were determined by the mesh or filter sizes. Otherwise, the dimensions of the imaging frame are used to specify the theoretical upper size range imaged by the device. ROI size estimates may include biovolume, area, or ellipsoidal axis for comparison. However, the size spectra for PSSdb were all

computed using ROI area for consistency across devices, since not all imaging instruments provide biovolume estimates and derived equivalent spherical diameter (see section 2.2.6 for more details). In addition, the value(s) for "Not Available" or NA were specified, if necessary, since we found some inconsistencies in the values reported, particularly for datasets generated by Zooprocess (i.e. UVP and scanner datasets), depending on the software version used, but also across variables for the same dataset. While the standardizer spreadsheet needs to be filled manually, we found this approach to be optimal to account for

the variable formats of existing and future datasets, both accessible online or directly sent to us.

Native units, defined in the standardizer spreadsheets, are converted to standard units using the python package Pint, designed to define, operate and manipulate physical quantities, based on units from the International System or defined in a custom text file. Custom units defined for PSSdb included the pixel per $\mu$m and pixel per mm used to convert pixel-based size estimates to metric-based estimates (https://github.com/jessluo/PSSdb/blob/main/scripts/units_def.txt). After standardization,

an interactive report is generated to check that units were correctly assigned by displaying the NBSS computed according to section 2.2.6, and the average particle size/concentration for individual samples. PSSdb developers can then check that both the size range and the overall concentration recorded are consistent with the particle size targeted by specific instruments (Lombard et al., 2019). This step ensures that file format and units in all datafiles are consistent, enabling the further merging of the data in the following PSSdb workflow steps.

### 2.2.4 Quality control of imaging datasets

After morphometric measurements, taxonomic annotations and metadata from the imaging data streams have been downloaded (see 2.2.2), the standardizer spreadsheets filled (see 2.2.3), and all datasets standardized, a quality control (QC) check is performed on individual IFCB, UVP, and scanner samples. The objective of this step is to ensure the good quality of the





datasets ingested in PSSdb, by automatically flagging individual samples whose size spectrum computation was either impossible (missing required information) or biased (incorrect GPS coordinates, pixel size, or low ROI number). We used a boolean factor to characterize each flag, assigning 0 (False) to non-flagged samples that passed the quality control and 1 (True) to flagged samples. Currently, 7 criteria are checked during the QC, and the overall flag is assigned 0 if the sum of the individual flags equals 0, and 1 otherwise.

The first flagging criterion stands for missing required data or metadata, as specified in the standardizer spreadsheets. Second and third, GPS coordinates are checked to verify whether they are located on land, according to the georeferenced Global Oceans and Seas dataset (v1 automatically downloaded from https://www.marineregions.org/), or located at 0x0° latitude/longitude, which sometimes indicates that this information has not been filled correctly. Fourth, to determine whether the number of ROIs ($n$) in a sample was sufficient to accurately estimate a size spectrum, we estimated count uncertainty assuming that particle detection followed a Poisson distribution (Schartau et al., 2010; Bisson et al., 2022). According to this distribution, the accuracy of ROI counts decreases significantly with lower count numbers $n$. We could thus estimate the probability of effectively observing $n$ ROIs given that the mean occurrence (the main parameter of the Poisson distribution) was equal to $n$, and assigned a flag to samples whose ROI counts yielded more than 5% uncertainty. Fifth, the percentage of manual taxonomic annotations (verified by a human expert) is calculated in order to flag samples that are less than 95% validated. This criteria is only applied to scanners and UVP datasets, as the larger number of IFCB images per sample make it more difficult to manually validate automated classifications. Sixth, the percentage of artefacts per sample is evaluated using the predicted or validated annotations so that any sample with 20% or more artefacts is flagged. Finally, samples with multiple pixel size factors are also flagged, since we do not expect the camera to be re-calibrated or replaced during deployment.

After the completion of the QC, a table summarising individual samples and their flags, along with an interactive report providing an overview of the samples flagged for each dataset, are automatically generated. The interactive report is checked by PSSdb developers and sent to the data providers, for an overview of the dataset sample locations, the number of ROIs, the percentages of validation/artefacts per sample, and the overall percentage of flagged samples. Hyperlinks are inserted in the interactive report to verify the sample information directly from the data source. Flags may be overruled by the data provider if they consider a sample is suitable (or not) for ingestion into PSSdb. For example, samples that have been size-fractionated could record a low ROI number, samples with a high percentage of artefacts may not necessarily be completely biased and low validation may be acceptable if all artefacts have been correctly identified.

### 2.2.5 Binning of imaging datasets

After standardization and QC, we first selected datasets where the sampling depth was between 0 and 250 m. Samples were aggregated spatially in 0.5x0.5° latitude/longitude cells, and temporally per week. This data aggregation approach allowed to (1) increase the overall volume analyzed per sample, which increases the number of particles observed and decreases the instrumental detection limit, and (2) avoid the over-representation of data from fixed time-series stations with high temporal sampling, compared to co-located "snapshot" samples in a given grid cell. The size spectra calculations described in the next section were performed on these weekly, 0.5x0.5° samples. Since individual weeks could occur in two separate months,




we assigned a unique month to each week by selecting the month that counted most samples. This approach prevented the creation of duplicate weekly samples per year. The final data products included in the first release (1a:bulk Normalized Bio-
volume/Abundance per size bin and 1b:slopes,intercepts, and determination coefficients of the size spectra) are reported as monthly, 1x1° grid averages, such that each mean size spectrum, slope, and intercept had a maximum sample size of 16, the product of four 0.5x0.5° sub cells in a 1x1° cell and four weeks per month. As mentioned above, reporting monthly, 1x1° grid parameter averages from the subgrid values, instead of calculating directly the size spectra for these larger bins, prevents a certain location/time series with a higher number of samples to skew the size spectra estimate, especially in a 1x1° cell that
contains both open-ocean sites sampled during research cruise(s) and coastal time series sites.

### 2.2.6    Computation of bulk particle size spectra and regression parameters from binned, instrument-specific datasets

The particle size classes used in PSSdb were previously defined in Kiko et al. (2022). These are logarithmically spaced using a base 2 and an increment of 1/3, so that a doubling in equivalent circular diameter (ECD) is observed every third bin (equivalent to a doubling in biovolume observed every bin), and range between 1-50,000,000 $\mu$m. The diameter of each particle,
with the exception of artefacts which are excluded from the size spectra computation, was estimated using area according to Eq. (1), and then converted to biovolume assuming a spherical shape of that diameter following Eq. (2).

$$\text{ECD} = 2 * \sqrt{\frac{\text{area}}{\pi}}$$
(1)

$$\text{Biovolume} = \frac{1}{6} * \pi * \text{ECD}^3$$
(2)

Area-based biovolume, rather than the more widely used distance-map estimates for IFCB datasets (Moberg and Sosik,
2012; Dubois et al., 2022), and ellipsoidal fits for scanners and UVP datasets, was selected to keep the size spectra calculations consistent across instruments. However, a sensitivity analysis of the slopes and intercepts as a function of the different size proxies (ellipsoidal, distance map, and area-based biovolume) is presented in (Fig. A2). Despite some differences in size spectra thresholding, likely due to elongated particles being assigned to different size classes (Fig. A2 a, b & c), our sensitivity analysis does not show any substantial differences in size spectra parameters from different biovolume estimates (Fig. A2 d, e
& f). This aligns with previous comparisons of elliptical or spherical biovolume derived size spectra which found no or little statistical difference between these estimates (Vandromme et al., 2012; Dubois et al., 2022).

The database includes size spectra calculated by two widely used methods: the Normalized Biovolume Size Spectrum (NBSS), routinely reported in zooplankton studies (e.g., San Martin et al., 2006), and the Particle Size Distribution (PSD), calculated from particle counters (broadly) or derived from satellite algorithms (Kostadinov et al., 2009; Kiko et al., 2022). For
NBSS, the Normalized Biovolume (NB) ($\mu$m$^3$ L$^{-1}$ $\mu$m$^{-3}$) for each biovolume size class (i) in a sample (0.5x0.5° grid cell, grouped by week) was calculated as the summed biovolume ($\mu$m$^3$), normalized by the cumulative volume sampled (L) and the Biovolume bin width ($\mu$m$^3$) as in Eq. (3):



$$NB_i = \frac{\dfrac{\sum Biovolume_{[i:i+1]}}{volume\ sampled}}{Biovolume\ bin\ width_i} \tag{3}$$

For PSD, the Normalized Abundance (NA) (number of particles $L^{-1}\ \mu m^{-1}$) for each size class (i) in a sample was

calculated as the total number of particles in ECD size class i, normalized by the cumulative volume sampled (L) and the ECD bin width ($\mu m$) as in Eq. (4):

$$NA_i = \frac{\dfrac{\sum particle\ count_{[i:i+1]}}{volume\ sampled}}{ECD\ bin\ width_i} \tag{4}$$

Retrieved size spectra were generally biased at the lower and upper size limits (Fig. 1e). At the lower end, the main bias is due to the sampling collection method (e.g., mesh of the net) or the segmentation threshold (e.g., minimum area or mean grey

level), which randomly excludes small particles, such that the closer the particles are to the camera resolution, the less likely they are to be imaged and segmented. At the higher end, imaging systems over-estimate larger, rarer particles whose concentration is close to the instrument detection limit, as determined by the imaging volume. As a result, size spectra would typically display an inflection point at the lower size limit and remain quasi-constant (e.g., flatter) at the upper size limit. The unbiased portion of the size spectrum was identified before computing the size spectra slopes and intercepts by log-linear regression.

To do so, we first exclude data from size classes with either a size measurement or particle count uncertainty greater than 20%, assuming Gaussian and Poisson error distributions, respectively. These distributions are based on the statistical analysis developed by Schartau et al. (2010) to quantify the size spectrum uncertainties, which assumes that size measurement uncertainties follow a Gaussian distribution with a variance equal to the camera resolution, and that the uncertainty of effectively observing ROIs given a similar occurrence of particles within the volume sampled follow a Poisson distribution. According to

this distribution, counting four or less ROIs in each size bin would yield an uncertainty greater than 20%. We thus reset the normalized biovolume/ abundance of size classes with four or less ROIs, mainly larger size classes, as empty size classes and selected the upper size limit as the largest size class before observing three consecutive empty size classes. Our choice for the upper size limit definition was a compromise between unnecessarily excluding large organisms and including too many large bin values that would bias the size spectra calculation towards flatter slopes. Next, we selected the size bin of the maximum

normalized biovolume/abundance value as the lower size limit. After selection, size spectra followed a power-law function in the form of Eq. (5), with a log-transformation resulting in a linear equation of the form described in Eq. (6):

$$NB_i = I\,(Biovolume_i)^b\,;\,NA_i = I\,(ECD_i)^b \tag{5}$$

$$\log_{10}(NB_i) = \log_{10}(I) + b*\log_{10}(Biovolume_i)\,;\,\log_{10}(NA_i) = \log_{10}(I) + b*\log_{10}(ECD_i) \tag{6}$$





The slope (b, $L^{-1}\ \mu m^{-3}$ for NBSS and $L^{-1}\ \mu m^{-1}$ for PSD), intercept (I, $\mu m^3\ L^{-1}\ \mu m^{-3}$ for NBSS and $\#\ L^{-1}\ \mu m^{-1}$

for PSD) and the coefficient of determination ($R^2$) of the size spectra were computed by log-linear regression following Eq.

(6). An easy way to interpret the intercept values is that it refers to the normalized biovolume and abundance predicted for a

standard 1 $\mu m^3$ and 1 $\mu m$ particle, respectively.

## 3   Results

### 3.1   Spatio-temporal coverage of imaging datasets

Up to 92,472 individual samples are included in the first release of PSSdb, which benefited from long-term IFCB time-

series collected at a 20 min frequency (Table 1). In comparison, the UVP and scanner datasets comprise fewer profiles/nets,

with a total of 3,068 profiles and 2,411 net samples, respectively.

**Table 1.** Spatio-temporal range of instrument-specific datasets included in the first release of PSSdb

| Instrument | No. Samples | No. 1°x1° spatial bins | Latitudinal range (°N) | Temporal range (years) |
|------------|-------------|------------------------|------------------------|------------------------|
| IFCB | 92,472 | 292 | 35–80 | 2013–2022 |
| Scanner | 2,411 | 169 | -65–81 | 1996–2022 |
| UVP | 3,068 | 861 | -65–80 | 2008–2021 |

These datasets span all major ocean basins, although most oceans are undersampled in the southern hemisphere. IFCB

datasets were all restricted to the mid- to high-latitudes of the northern hemisphere (Fig. 2, Table 1). Further, the majority of

IFCB samples are located on the shelf of the Eastern and Western United States, due to the presence of long-term time-series

sites of the California Ocean Observing System and the Northeast U.S. Shelf Long-Term Ecological Research programs (Fig.

2a). UVP and scanner datasets are distributed more evenly across the ocean basins, mostly due to the Tara Ocean (2009-2012)

and Tara Polar Circle (2013) global expeditions, even though specific monitoring programs increased the density of samples in

the tropical Atlantic, the eastern temperate Atlantic and the Mediterranean Sea (Fig. 2b,c). These monitoring programs resulted

in a large temporal coverage of the three instrument-specific datasets, with repeated observations sustained for periods of 10-

25 years (Fig. 2d,e,f; Table 1). Notably, the scanners show the largest temporal coverage, from 1996 to 2022, by including

samples collected at the long-term monitoring sites located in the Bay of Villefranche-sur-mer and the Bay of Biscay (France).

The gap observed between 1998 and 2003 is caused by the exclusion of samples that had not been validated to at least 95%.

This high-frequency dataset affected the monthly variability of scanners sample density, shown in Fig. 2e, since the Bay of

Biscay monitoring program only takes place in May (Grandrémy et al., 2023a). UVP datasets have the second longest coverage

with observations collected between 2008-2021 (Fig. 2f). In PSSdb first release, the climatology of UVP sampling density is

slightly biased towards spring months (March, April and May), however, this may not reflect actual sampling efforts, as UVP

images also need to be more than 95% validated to be ingested in PSSdb. This threshold is not applied to IFCB datasets, which

comprise too many images to be manually curated, yet the datasets also show a strong bias towards the summer months (June,

July, August). This bias reflects the sampling strategy of both the NESLTER broadscale, limited to the summer months, and CalOOS sampling programs, which partially operate with IFCBs serviced during the wintertime to avoid damage. IFCB has been routinely deployed at the Martha's Vineyard Coastal Observatory since 2006, however only samples from 2013 and after were included in PSSdb as previous observations did not include taxonomic predictions, which were required to filter artefacts out of the data products (Table 1; Fig 2d).

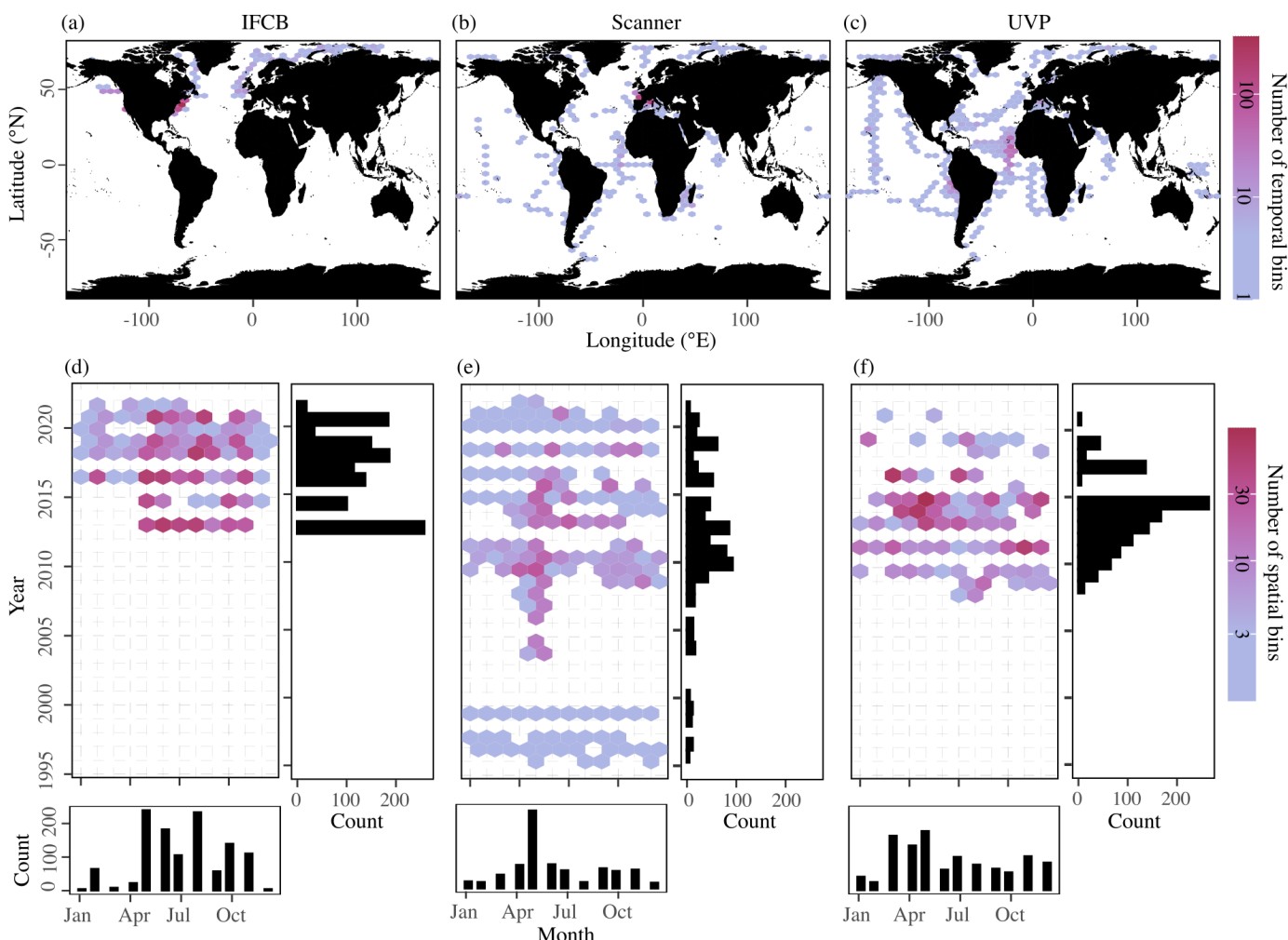

**Figure 2.** Spatio-temporal coverage of PSSdb first release datasets obtained from IFCB (a,d), scanner (b,e), and UVP (c,f). Maps and hovmöller diagrams are color-coded according to the density of temporal bins (top panels), corresponding to year and month, and spatial bins (bottom panels), corresponding to 1x1° grid cells, respectively. The size of the grid cells are expanded (∼x2) in panels a,b and c to help visualize the color scale, and represent a coarser spatial coverage of the dataset.



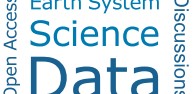

**Table 2.** Size spectra description for each instrument included in the first release of PSSdb. Parameters are reported as mean ($\pm$ standard deviation), with the exception of the untranslated intercept, which is reported as a geometric mean, with the range of observed values given the first and third quartiles in the parentheses.

| Instrument | ROI size range | Proxy | Slope | Intercept | | $R^2$ |
|---|---|---|---|---|---|---|
| | ($\mu$m) | NBSS | $L^{-1}\,\mu m^{-3}$ | $\log_{10}(\mu m^3\,L^{-1}\,\mu m^{-3})$ | $\mu m^3\,L^{-1}\,\mu m^{-3}$ | |
| | | PSD | $L^{-1}\,\mu m^{-1}$ | $\log_{10}(\text{particles } L^{-1}\,\mu m^{-1})$ | particles $L^{-1}\,\mu m^{-1}$ | |
| IFCB | 2.88-369.1 | NBSS | -0.95($\pm$ 0.30) | 7.3($\pm$ 1.2) | $2\times10^7$ ($5\times10^6$,$6\times10^7$) | 0.92($\pm$ 0.08) |
| | | PSD | -3.17($\pm$ 0.91) | 7.5($\pm$1.27 ) | $3\times10^7$ ($6\times10^6$,$1\times10^8$) | 0.93($\pm$ 0.06) |
| Scanner | 116.4-26,995 | NBSS | -0.99($\pm$ 0.25) | 7.5($\pm$ 2.2) | $3\times10^7$ ($1.5\times10^6$,$4\times10^8$) | 0.94($\pm$ 0.06) |
| | | PSD | -3.30($\pm$ 0.75) | 7.7($\pm$ 2.3) | $5\times10^7$ ($2\times10^6$,$6\times10^8$) | 0.95($\pm$ 0.04) |
| UVP | 92.5-15,272 | NBSS | -1.11($\pm$ 0.22) | 8.9($\pm$ 1.7) | $8\times10^8$ ($1.2\times10^6$,$8\times10^{11}$) | 0.96($\pm$ 0.03) |
| | | PSD | -3.65($\pm$ 0.65) | 9.2($\pm$ 1.7) | $1\times10^9$ ($7\times10^7$,$1\times10^{10}$) | 0.97($\pm$ 0.02) |

## 3.2 Size spectra obtained from individual imaging devices

The mean values of the size spectra derived parameters (i.e., regression slope, intercept, and determination coefficient, or $R^2$) per instrument, as well as the effective size range sampled by each instrument are reported in Table 2. The IFCB effectively detects and images plankton and detrital particles in the nano (2-20 $\mu$m) and micro (20-200 $\mu$m) size fractions. This size range is supplemented by UVP and scanner datasets, which include predominantly living microplankton (20-200 $\mu$m), mesoplankton (200-2,000 $\mu$m) and macroplankton (2,000-20,000 $\mu$m). The UVP additionally samples fragile organisms and non-living particles, which are disrupted by the net collection (e.g., Biard et al., 2016; Soviadan et al., 2023). As a result, the UVP-specific spectra were consistently higher than the scanner spectra.

We used two metrics to evaluate pelagic size structure from plankton imaging devices: the NBSS, computed with normalized biovolume (Eq. 3), and the PSD, computed with normalized abundance (Eq. 4). Both metrics showed similar patterns, resulting in high correlations between the fitted parameters (Fig3a,b), namely the NBSS and PSD slopes (r=0.99), intercepts (r=0.99), and determination coefficient $R^2$ (r=0.98) (Fig3c,d). For simplicity, we further describe observed patterns of the instrument-specific size spectra parameters derived from NBSS only, since both PSD and NBSS co-vary. However, all patterns and trends described in the following sections, including in the discussion, hold for the PSD releases.

Global size spectra slopes and intercepts were relatively consistent between instruments, with average values of $\sim$ -1 $L^{-1}\mu m^{-3}$ and $\sim$ 7.9 $\mu m^3\,L^{-1}\,\mu m^{-3}$ (corresponding to an approximate concentration of $8\times10^7$ $\mu m^3\,L^{-1}\,\mu m^{-3}$ for particles of 1 $\mu m^3$) respectively (Table 2, Fig. 3). UVP's size spectra presented an intercept slightly above that of the IFCB and scanners, given the additional particles they can detect *in situ*, with overall higher $R^2$ estimates, although relatively large $R^2$ were observed across all instruments (Table 2, Fig. 3).





**Figure 3.** First release of PSSdb: Pelagic size spectra (products 1a) approximated from normalized biovolume (a) and normalized abundance (b), and comparison between fitted (products 1b) slopes (c) and intercepts (d) for the three plankton imaging systems included in the first release. Solid lines in panels (a) and (b) represent the median spectrum, restricted to size classes that were observed in at least 50% of the samples, to avoid misalignment due to different sampling efforts (e.g. different mesh sizes for scanners, different PMT settings for IFCB).





### 3.3 Size spectra regression fit, slopes and intercepts

In addition to the average, instrument-specific differences reported above, PSSdb allows exploration of the spatial and temporal variation in the NBSS and PSD (not shown since they co-vary with NBSS) for individual instruments. Fig. 4 shows the average NBSS slopes, intercepts and $R^2$ obtained for each grid cell in the global ocean. Despite their similar size targets, there were substantial differences in the global distribution of NBSS slopes and intercepts derived from the three imaging approaches. Indeed, while the majority of the slopes were around -1 $L^{-1}$ $\mu m^{-3}$, the scanner slopes showed no clear variation

with space. Meanwhile, the UVP slopes tended to be low (i.e steeper size spectra) within oligotrophic gyres and higher (i.e flatter size spectra) in the northernmost latitudes or by the coasts (Fig. 4c). This pattern was inverted with regards to the intercepts, as the abundance of 1 $\mu m^3$ particles was lower in the Arctic and increased near shore (Fig. 4f). Likewise, the IFCB NBSS slopes were higher, and intercepts lower, in the northernmost latitudes and along the Eastern coast of the United States, compared to the Western coast (Fig. 4a,d). The determination coefficients seemed to follow an inverse relationship with the

slope for IFCB NBSS, as flatter NBSS were also marked by lower $R^2$ (Fig. 4g). The scanner data however, did not follow such clear trends and seemed less variable than the UVP and IFCB (Fig. 4b,e), although there seemed to be a clear decrease of NBSS linearity, or $R^2$, towards the pole (Fig. 4h).

       To check whether these trends were specifically linked to sampled latitude, we looked at the latitudinal variability of the NBSS parameters (Fig. 5). IFCB measurements were all restricted to a small latitudinal range, however, we observed a

notable decrease in the linearity of the size spectra with latitude. Higher latitudes (>50°N and S) also showed higher variation in both slope and intercept estimates compared to lower latitudes, as well as lower coefficients of determination for scanner and UVP size spectra. Both show a reduced variability in derived slopes and intercepts within the tropics, with flatter slopes and increased intercepts notably located at 0°N, nearby the Equatorial current system. Since latitudinal trends can be impacted by different dynamics in specific regions, but also by differences across seasons, we computed the instrument-specific monthly

climatologies of NBSS parameters in ocean regions where there was at least 10 months of data (Fig. 6). This excludes the Arctic Ocean, Red Sea, South Atlantic, Southern Ocean and Baltic Sea, which are represented in PSSdb, but do not have enough data to resolve seasonal cycles.

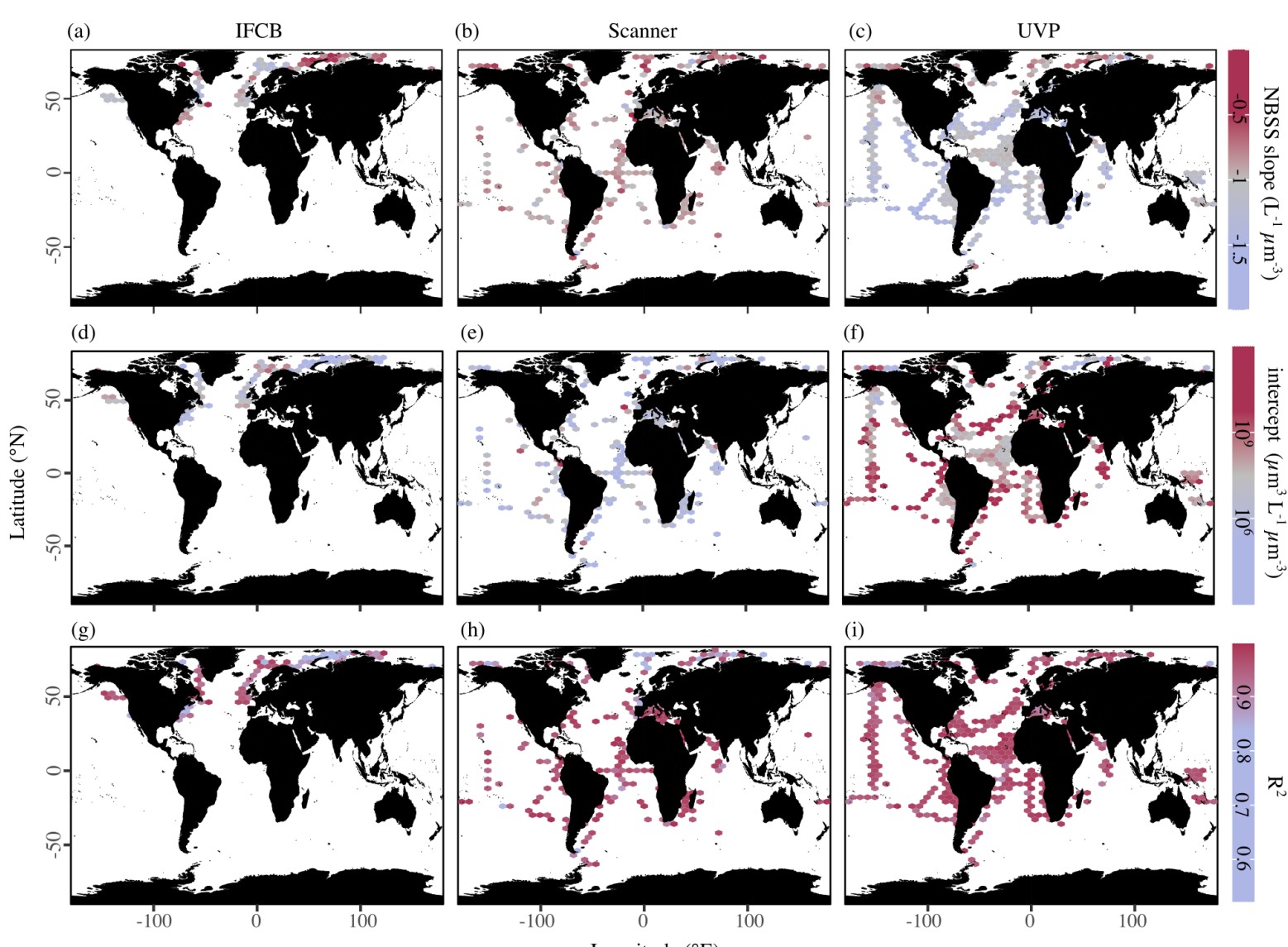

**Figure 4.** Average NBSS parameters in 1x1° latitude/longitude cells (products 1b), from imaging data obtained by IFCB (a,d,g), scanners (b,e,h), and UVP (c,f,i). Slopes correspond to panels (a,b,c), intercepts to panels (d,e,f) and determination coefficients to panels (g,h,i). The size of the grid cells are expanded (∼x2) in all panels to help visualize the color scale, and represent a coarser spatial coverage of the dataset.

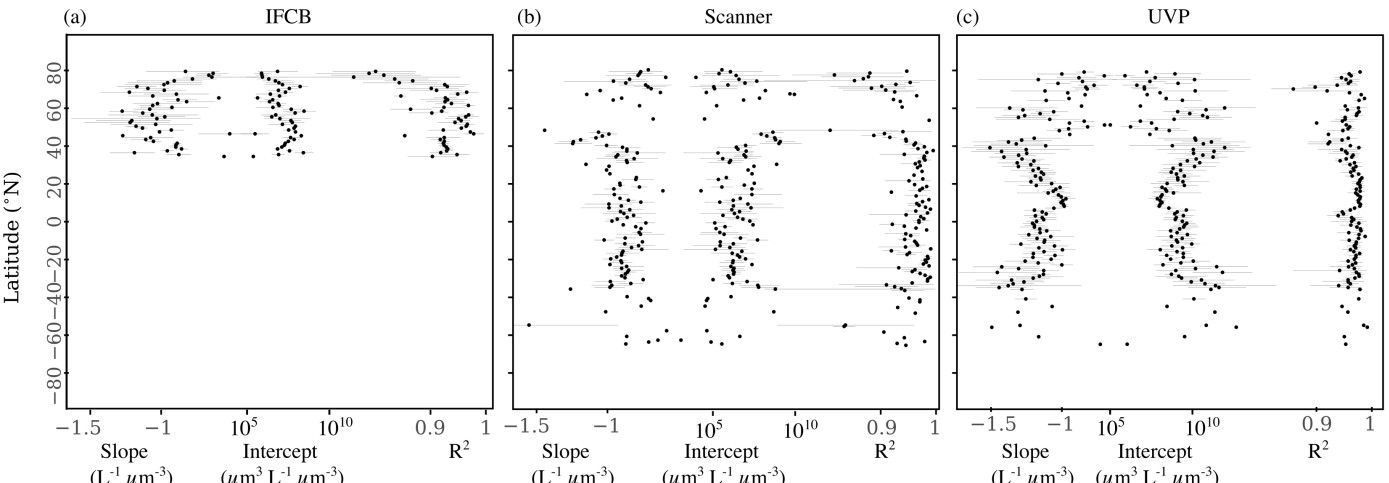

**Figure 5.** Latitudinal variability of NBSS slopes, intercepts and determination coefficients for IFCB (a), scanners (b), and UVP (c). Dots represent the mean parameter value per 1° latitudinal bins, and the horizontal bars represent the standard deviation.

Time series analysis of the instrument-specific NBSS showed pronounced seasonal cycles, but high variability by region, and in some cases, between instruments. Seasonal variations in NBSS parameters were apparent for most ocean basins, as well as in the Mediterranean Sea (Romagnan, 2013), which showed high variation of scanner mean slope and intercept through the year (Fig. 6b,g), with rather constant $R^2$ values throughout the year (Fig. 6l). Stable $R^2$ values were generally observed across all instruments and ocean basins, with the exception of the scanner datasets located in the Indian Ocean, which presented a large dip in NBSS linearity in October (Fig. 6k). Interestingly, the North Atlantic presented opposite trends between the IFCB and scanner, whose NBSS slopes decreased (i.e., steepened) and intercepts increased during the spring (Fig. 6c,h), and the UVP datasets, where NBSS slopes increased (i.e., flattened) and intercepts decreased during spring and summer (Fig. 6c,h). In the southern hemisphere, UVP slopes were at a minimum by the end of austral summer (January-May), with a concurrent increase in NBSS intercepts only observed by the end of this period (Fig. 6e,j). Scanner datasets showed similar trends to that of the UVP in the South Pacific, except that the minimum slope and maximum intercept were observed by March, earlier in the year. The Indian Ocean followed the same seasonal cycle, with large differences between seasons as slopes decreased (i.e., steepened) and intercepts increased during the spring-summer transition, and remained relatively stable at high slope and low intercept values from September through November (Fig. 6a,f). Lastly, the IFCB datasets collected in the North Pacific presented two peaks in NBSS intercepts, with concurrent dips of slopes indicative of steeper NBSS, by spring (April) and fall (Oct) (Fig. 6d,i).



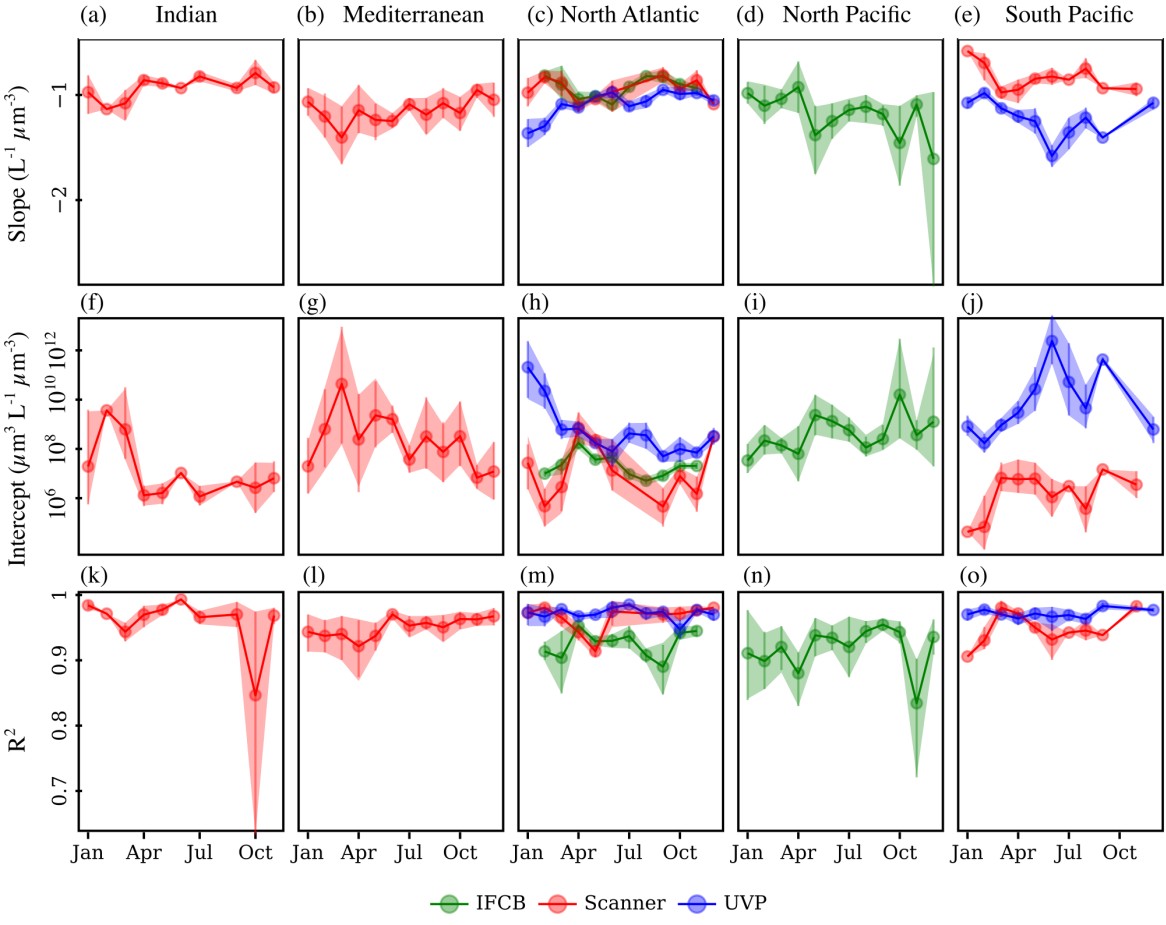

**Figure 6.** Climatologies of NBSS slopes (a,b,c,d,e), intercepts (f,g,h,i,j) and $R^2$ (k,l,m,n,o) for each imaging system. The data is shown for five major ocean basins with enough data to show seasonal fluctuations: Indian Ocean (a,f,k), Mediterranean sea (b,g,l), North Atlantic (c,h,m), North Pacific (d,i,n) and South Pacific (e,j,o). Vertical lines represent the standard deviation of the monthly average parameters.

Given the opposite spatial (Fig. 4, 5) and temporal (Fig. 6) patterns often observed between the size spectra slopes and
intercepts, across instruments and oceanic regions, we used the yearly time series correlation of these 2 parameters in any given grid cell within the same oceanic region as a way to detect potential decoupling, lag, or feedback between the two. The (de)-coupling between NBSS slopes, which represent the balance between relatively small and large particle and plankton, and intercepts, which approximates the carrying capacity of a given ecosystem, across the years is presented in Figure 7. As expected, the majority of PSSdb grid cells were strongly anti-correlated, with coefficients close to -1, since lower slopes
(e.g., steeper size spectra) tend to indicate an increased proportion of smaller particles, which are generally more abundant. Noticeably though, there are also areas of strong positive relationship between the two parameters, especially within the IFCB datasets located in the North Atlantic. Flatter NBSS were thus associated with increased abundances of 1 $\mu m^3$ organisms, which





could be indicative of the relief of nutrient stress allowing for multiple phytoplankton size groups to co-exist (Armstrong and McGehee, 1980), other complex interactions between primary producers dictated by resource competition, or trophic shunt between small and large plankton for zooplankton. In this region, we also observed a de-coupling between the NBSS parameters for 2-3 years, as indicated by grid cells with low absolute correlation coefficients. A de-coupling between size spectra parameters could arise from temporal lag in trophic transfer and complex trophic cascading, similar the one mentioned above. Care should be taken when testing for significant long-term trends in the coupling of the NBSS parameters and detecting yearly perturbations, however we expect such analysis to become more robust as more datasets are ingested into the future releases of PSSdb.

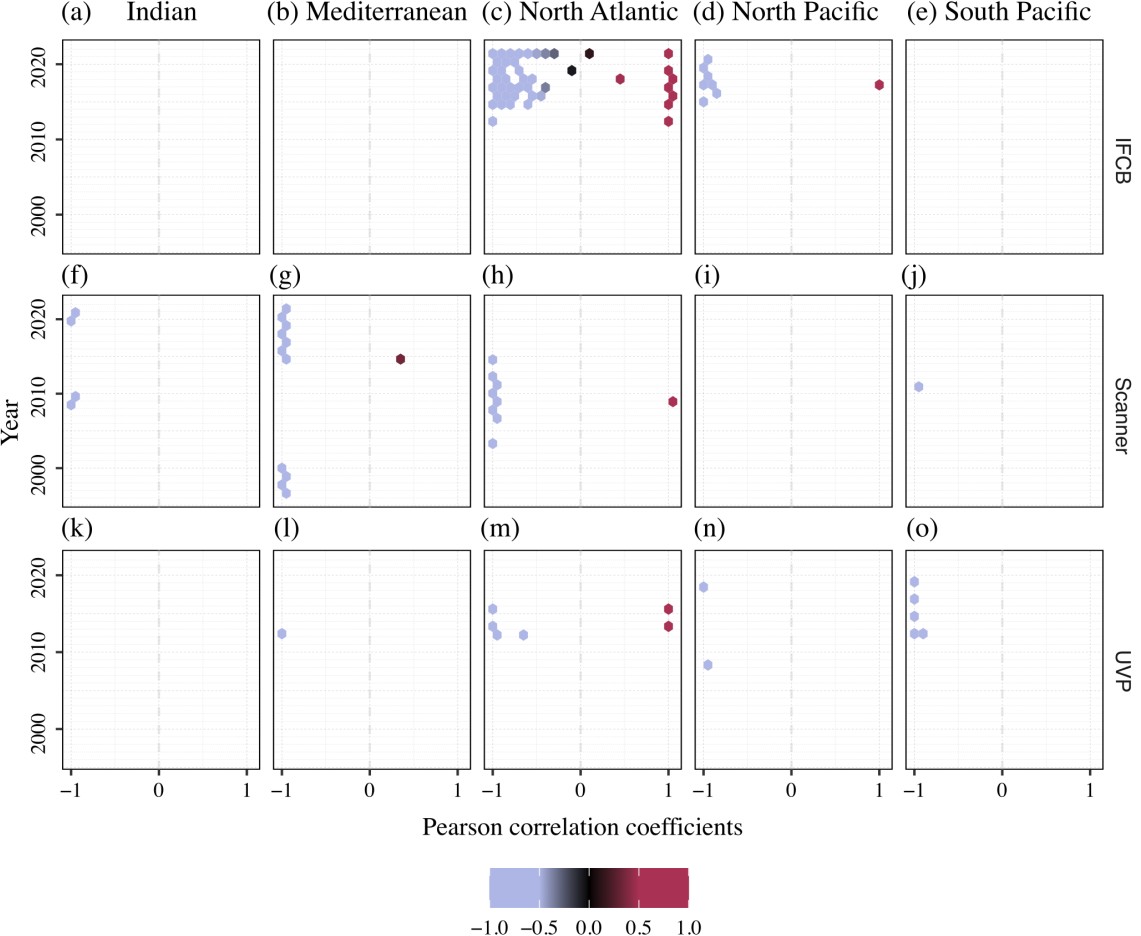

**Figure 7.** Pearson correlation coefficients between NBSS slopes and intercepts across ocean basins (columns) and years for IFCB (a,b,c,d,e), Scanner (f,g,h,i,j), and UVP (k,l,m,n,o). Each datapoint represents an individual grid cell within the five major ocean basins with enough data to show seasonal fluctuations: Indian Ocean (a), Mediterranean sea (b), North Atlantic (c), North Pacific (d) and South Pacific (e).



## 4 Discussion

Workflows that provide estimates of plankton size distributions with an extensive spatial and temporal coverage will greatly accelerate efforts to characterize and understand ecological plankton dynamics at a global scale. With this goal, the first PSSdb data products were generated to determine patterns in trophic transfer efficiency across plankton sizes and ecosystem
carrying capacity, through consistent measurements of particle sizes analyzed by three state-of-the-art plankton imaging devices. In this section, we discuss how the spatial and temporal coverage of these instrument-specific datasets effectively reduce the gap of available size structure observations, before presenting potential uses of the datasets and future directions for the database.

### 4.1 The contribution of PSSdb and other data compilations in reducing the gap of available size structure estimates

A global compilation of particle size distribution has been published recently by Kiko et al. (2022), using the UVP5 bulk particle size distribution accessible from EcoPart. Other recent studies (e.g., Hatton et al., 2021) have also constructed global estimates of size distribution in marine organisms using indirect biomass estimates of arbitrary plankton size classes derived from satellite proxies, models,or data compilations like COPEPOD (Moriarty and O'Brien, 2013) and the MARine Ecosystem biomass DATa (MAREDAT, Buitenhuis et al., 2013), without relying on direct size estimates. Such databases and compilation
efforts have benefited from exponentially growing sampling efforts during the past decades, with hundreds to thousands of new UVP profiles generated each year (Kiko et al., 2022). Yet, to our knowledge, our workflow is the first attempt to compile the counts, size measurements, and taxonomic information of individual particles from multiple imaging devices to generate global particle and planktonic size spectra datasets, aimed to be accessible to the broad scientific community. Similar to the COPEPOD database, we have focused our effort on compiling data from different instruments, sampling regimes, and data
collection methodologies in a self-consistent and cross-calibrated manner, enabling ease of comparison between all the major ocean basins and across sampling systems.

So far, size spectrum studies have been restricted to accessible areas and clement weather conditions (Hatton et al., 2021), leading to fewer sampling efforts at high latitudes, specifically the Southern Ocean and the South Pacific Ocean. Similarly, the sampling density is skewed towards the continental shelves, as opposed to open ocean stations. Like other global compilations,
our datasets of marine pelagic size structure highlight multiple undersampled regions by plankton imaging systems. All imaging sensors were mostly deployed in the Northern hemisphere, in contrast with fewer deployments in the South Pacific, the Western Indian Ocean, and the Southern Ocean. While the latter is covered in the UVP-based compilation of particle size structure (Kiko et al., 2022), the absence of the Southern Ocean in our database results from the need for the manual validation of taxonomic annotations to pass our current quality control. However, as more autonomous vehicles will be equipped with UVP6 and their
embedded classifier (Ricour, 2023), notably in the Bio-Argo program (Claustre et al., 2020), we anticipate that UVP6-derived datasets will grow substantially in the upcoming decade. To accommodate the growth of datasets derived from large-scale surveys, we could relax such criteria to generate specific data products in near-real time. Since a few UVP6 datasets are already incorporated in this initial release, we expect further ingestion of additional UVP6 data to be relatively straightforward.





Unlike UVP datasets, IFCB and scanner datasets are more difficult to compile, due to the lack of a common platform
to manage incoming datasets, and increased efforts needed during the sampling (e.g., net deployments and recoveries), pre-processing (e.g., concentration or size-fractionation of particles before imaging) and post-processing steps. Notably, the number of images collected at hourly/sub-hourly frequency by IFCB devices and their classification is a veritable bottleneck to produce near real-time datasets. Only 16% of accessible datasets found in our IFCB data sources were ingested in PSSdb due to the lack of taxonomic predictions (needed for detection of artifacts), although we expect more datasets to provide such predictions
soon with the rapid development of new classifiers. Our datasets were thus restricted to the Northern latitudes, highlighting a need for future sampling efforts targeting the Southern hemisphere like BIO-GO-SHIP (Clayton et al., 2022). To overcome this limitation, user-specific settings that trigger the image acquisition based on a specific size/fluorescence value may help reduce the total number of images to classify and the presence of smaller cells (4-7 $\mu$m) that are harder to identify even manually. Alternatively, newer, more efficient, automated classifiers can also help manage upcoming observations (Kraft et al., 2022).

**4.2  Global patterns and trends in plankton size spectrum: insights from PSSdb first release and potential future uses**

Plankton and particle size spectrum derived parameters (slope, intercept, and determination coefficients) are all important indicators of ecological processes (Sprules and Munawar, 1986; Trudnowska et al., 2021). As such, they can inform us on the general functioning and state of pelagic ecosystems, and eventual perturbations or shifts in plankton community structure.

The compilations from Hatton et al. (2021) and Kiko et al. (2022) both seem to support the presence of an equal stock of
living biomass across increasing size classes (slope of the biomass spectrum equals to ∼0), driven by the log-linear decline of particle abundance with increasing size/biomass (slope of the normalized biovolume/abundance spectrum equals to ∼-1 and -4 respectively), postulated by Sheldon et al. (1972). Like these studies, the majority of PSSdb NBSS slopes are relatively close to -1 (equivalent -4 for PSD), indicating a stable equilibrium between small and large particles and a similar trophic transfer efficiency (Fig. 4, Table 2). Nevertheless, substantial divergence from the canonical slope were observed for all the instruments
used in this release, notably in the northernmost latitudes and close to the coasts. Size spectra slopes have been shown to increase (less negative or flatter slope) with increasing nutrient supply (e.g., upwelling, coastal, and polar systems), as observed by other data compilations (see Atkinson et al., 2021, for freshwater ecosystems), modeled from size-structured plankton systems (Barton et al., 2013; Hatton et al., 2021; Serra-Pompei et al., 2022) or approximated from satellite data (Kostadinov et al., 2009; Hirata et al., 2011; Roy et al., 2013). Interestingly, we did not observe increased slopes in stable upwelling
ecosystems, located by the Californian, Peruvian, Namibian and northwestern African coasts (Fig. 4). The shallowing of size spectra slopes with increasing nutrient supply is not a universal pattern, since flatter size spectra have also been reported in stable, oligotrophic ecosystems, compared to more productive ecosystems (Marcolin et al., 2013; Atkinson et al., 2021). The former are typically considered at steady-state, as reflected in the stable daily oscillations of total particulate organic carbon, yet significant variability in time and space raises substantial concerns regarding our ability to extrapolate plankton size spectra
and their slopes from crude or fragmented observations (Rodriguez and Mullin, 1986).

A simple explanation for this lack of consistency is that all spatial patterns are effectively impacted by sampling timing. Notably, our extended temporal coverage in the Indian, Pacific, Atlantic Oceans, as well as the Mediterranean Sea, have high-





lighted that there is significant variability in size spectra slopes and intercepts, from month to month (Fig. 6). Most temperate regions presented a trend consistent with the formation of a spring bloom, indicated by a flattening of the size spectra, and its
progression towards a more stratified environment, marked by steeper size spectra due to the predominance of smaller plankton, in agreement with other regional and global studies (Clements et al., 2022; Haëntjens et al., 2022). However, coastal regions sampled by the IFCB showed an opposite progression with steeper size spectra during the spring and fall seasons, consistent with a shift of the phytoplankton community towards smaller dinoflagellates, compared to larger diatom chains, as described in Fischer et al. (2020). Such shifts should be detected early, through comparison with longer time periods provided by PSSdb
data products, and monitored in time, especially if linked to harmful algal blooms that represent an important threat to human health around the globe (Glibert, 2020). In this case-study, the appearance of small dinoflagellates was also linked to a lower coefficient of determination. This parameter decreases with the non-linearity of particle size spectra, and as such can be an important indicator of ecosystem perturbations and non steady-state conditions.

Most studies assessing marine plankton size structure have focused largely on analyzing the slope, and to a lesser extent
the intercept of pelagic size spectra, with much less interest given to the coefficient of determination ($R^2$). However, differences in size spectrum linearity can arise from abiotic or biotic perturbations leading to local peak(s) of intermediate-size organisms (Moscoso et al., 2022). "Bumps" in the plankton size spectrum have been reported or modelled under harmful algal blooms (Harred and Campbell, 2014), transient trophic interactions (Schartau et al., 2010; Banas, 2011; Rossberg et al., 2019), and as the result of mesoscale circulation (Noyon et al., 2022) or the omission of specific groups in the observed size range (e.g.,
heterotrophic nanoflagellates not detected by most imaging flow cytometers targeting fluorescing organisms, see Chisholm, 1992). Non steady-state conditions are increasingly observed, particularly in nutrient-rich systems (Cavender-Bares et al., 2001), and represent a considerable interest for environmental policies. For this reason, we carefully assessed and reported size spectra non-linearity in our database, along with the other, widely analyzed, parameters. Our first-release products show that regions with lower $R^2$ were mostly located toward the North Pole, and were particularly linked to lower (e.g., flatter) size
spectra slopes in these regions (Fig. 4, 5). Like a lower $R^2$, a decoupling between size spectra parameters is also indicative of important perturbations, or inversely of the resilience, of a given ecosystem via complex trophic interactions (e.g., temporal lag, resource competition, grazing cascades). We suggest to follow the yearly correlation between slopes and intercepts, as presented in Fig. 7, to detect potential deviation from the expected seasonal trends, showing anti-correlation between size spectrum slopes and intercepts (Fig. 6). More data will greatly improve the accuracy of such analysis, and potentially help
inform policy stakeholders by revealing significant, climate-driven trends in size spectra decoupling.

A more detailed interpretation of our observed patterns and trends is out of the scope of this manuscript. However, we hope PSSdb will be further exploited by individual research groups or stakeholders to contextualize their study or policies. In addition, current modelled (Serra-Pompei et al., 2022) and satellite-derived (Hirata et al., 2011; Roy et al., 2013; Kostadinov et al., 2023) plankton size distribution have yet to be compared to extensive size structure observations. PSSdb could represent a
potential avenue to assess the performance of models and satellite proxies, especially as new and future model outputs (Negrete-García et al., 2022) and satellite datasets (PACE, https://pace.oceansciences.org/) will provide biomass measurements for an ever increasing number of plankton functional groups. Such validation is key to constraining some of their uncertainties, and



gain a mechanistic understanding of how physiological and ecological processes structure current and future marine ecosystems (Menden-Deuer et al., 2021). In addition, PSSdb users could investigate important factors driving the observed spatial patterns
and temporal trends of plankton size spectra. PSSdb products could thus improve our understanding of the temporal and spatial variability of particle size spectra in specific regions and provide a broader context to case studies, as showcased in Fig. 4 to 7, and support global data-driven interpolation, similar to Hatton et al. (2021) or Clements et al. (2022).

**4.3 PSSdb successes, challenges and further considerations to maintain and expand the database**

In our effort to access and compile imaging datasets from multiple devices, we found the open source platforms (and
associated APIs) developed for IFCB, UVP, and scanner users to manage their incoming datasets instrumental. For example, the online dashboards are a useful tool for IFCB data generators to assess image quality during and post-deployment, by quickly checking the raw images and monitoring the number of ROIs per sample, and alert potential stakeholders when a species of interest is detected. However, the possibility to link a set of metadata and a tag (e.g., in case of suspicion of any bias) for each sample was only added recently on second-generation dashboards. As a result, a significant number of datasets accessible on
first-generation IFCB dashboards were not ingested in this initial release. It is difficult to assess how many IFCB samples were not ingested due to such lack of metadata, as an exhaustive list of IFCB dashboards, that would enable better data traceability, is still missing. Similarly, a portion of scanner and other net-collected imaging datasets are not easily traceable or usable for PSSdb, as some data collectors still use early tools (Zooprocess and PlanktonIdentifier which is no longer supported) to manage their datasets. Even though our pipeline is able to ingest datasets directly sent to us, these datasets are eventually harder
to trace and compile compared to UVP datasets which are, to our knowledge, all uploaded on EcoTaxa and EcoPart. Both web platforms offer a secured, easy, and reproducible access to numerous datasets, and for EcoTaxa, to images annotations, a key feature to follow the status of the UVP and scanner datasets that should be validated to at least 95% to be ingested in PSSdb.

These open source management platforms have been available to the scientific community for a decade, but still suffer from a general lack of funding to support their development and maintenance. This contrasts with the increasing funding to
develop new imaging prototypes and commercial instruments (Lombard et al., 2019; Martin-Cabrera et al., 2022). Examples of imaging instruments that were not ingested in the PSSdb initial release include the Planktoscope (Pollina et al., 2022), the CytoSense (Dubelaar and Gerritzen, 2000), the FlowCam (Sieracki et al., 1998), the ZooGlider (Ohman et al., 2019), the ISIIS (Cowen and Guigand, 2008), the CPICS (Gallager, 2016), the VPR (Davis et al., 2005), and the LOKI (Schulz et al., 2010). From their associated publications, it is unclear how these datasets are archived in long-term repositories, although a
few datasets collected with Planktoscope, ZooCAM, CytoSense, and FlowCam instruments have already been uploaded on EcoTaxa. Ingesting such datasets in the PSSdb database would be extremely valuable to assess extended plankton size spectra in the millimeter-centimeter size range, and bridge some of the gaps introduced by specific instrument operational ranges while providing overlapping size bins (Haëntjens et al., 2022). The latter are key for pooling datasets obtained from multiple imaging devices deployed in spatial and temporal proximity. In some cases, merging imaging datasets integrated over specific depth
layers (e.g., net-collected datasets) with profiling or towed datasets is facilitated by simply integrating observations using the lowest sampling resolution (Soviadan et al., 2023); but merging discrete (e.g., surface-only) and integrated observations is





more problematic without a good understanding on how the discrete measurements might change with depth. Despite such challenges, the relatively small differences between the overall intercepts and slopes of PSSdb first release products is greatly encouraging (Table 2). Prior to PSSdb, efforts to set guidelines and best practices for obtaining plankton observations with
imaging instruments (see Lombard et al., 2019; Neeley et al., 2021) had yet to establish protocols on harmonizing these datasets across platforms, given the large variability between sampling strategies, instrument detection limits, size estimates, organisms targeted, and classification schemes. We hope to build upon this first data release and recent work from (Soviadan et al., 2023) to provide merged data products, that will effectively span the five orders of magnitude that can be captured by commercially available plankton imagers (Lombard et al., 2019).

Further, we are also planning on releasing taxonomically resolved PSSdb products, which will allow for the analysis of temporal and spatial shifts in plankton community composition, since individual size observations collected from imaging devices are mostly paired with taxonomic annotations. Thus, it will be possible to assess taxon-specific size spectra using the same pipeline that we developed for the raw particle products, with minor modifications. These future products will incorporate different levels of taxonomic resolution, allowing a global assessment of group-specific size structure and derived biomass based
on published relationships linking biovolume to carbon content (Menden-Deuer and Lessard, 2000; Lehette and Hernández-León, 2009; McConville et al., 2017). The lack of standardization across classification schemes and taxonomic experts will likely be a challenge, as they both lead to disparate ranking of taxonomic annotations across imaging datasets, which are harder to homogenize. In the future, fine taxonomic resolution could be achieved by following the recent guidelines and standards for image annotation published by Neeley et al. (2021). Such effort should be facilitated by the availability of extensive
training sets already published online for IFCB (https://hdl.handle.net/10.1575/1912/7341), ZooScan (https://www.seanoe.org/data/00446/55741/), and ISIIS (https://www.ncei.noaa.gov/access/metadata/landing-page/bin/iso?id=gov.noaa.nodc:0127422) images. Combined with newer classifiers (Kraft et al., 2022; Eerola et al., 2023), these could greatly accelerate the turnover for data processing and availability to reach operational plankton monitoring. More practically for the current heterogeneity of image classification schemes, annotations could be grouped into broad categories, like plankton functional groups used in
current ocean biogeochemical (OBGC) models.

## 5 Summary and conclusion

In this paper, we present a first compilation of pelagic size spectra obtained from three imaging systems: the IFCB, UVP and scanners. They represent state-of-the-art technologies to count, size, and identify living and non-living marine particles in the 7-10,000 $\mu$m size range, but their datasets had not been accessed, compiled, and shared in a consistent and interoperable
manner so far. To facilitate a global compilation of size observations obtained with imaging instruments and promote near-real time assessments of plankton size distributions, we thus developed an open-source pipeline, available at https://github.com/jessluo/PSSdb. Using this pipeline, we gathered hundreds of specific datasets spanning most of the global Ocean, with the exception of the Southern Ocean and South Pacific.





Our first-release products, available at https://doi.org/10.5281/zenodo.10150020, show consistent decline of raw parti-
cle numbers with increasing sizes across the 7-10,000 $\mu$m size range, with a slope close to -1 L$^{-1}\mu$m$^{-3}$ (for NBSS and -4
L$^{-1}\mu$m$^{-1}$ for PSD), in agreement with other size structure compilations, and an average intercept of 8x10$^7$ $\mu$m$^3$ L$^{-1}\mu$m$^{-3}$.
Substantial divergences were observed in space and time for both parameters, which could point toward changes in trophic ef-
ficiency and overall carrying capacity of marine ecosystems, especially in regions of increased nutrient supply. Those changes
were sometimes linked to a change in size spectrum linearity and in the coupling between size spectra parameters, which can
be driven by specific processes and perturbations such as blooms. Targeted analysis of the spatio-temporal variations and per-
turbations of the plankton size spectra will improve our understanding of important processes and feedback governing marine
ecosystems, and help constrain the uncertainty around future projections of marine diversity, services, and biogeochemistry
from data-driven and mechanistic models.

We plan on adding datasets to PSSdb and to this end, encourage all research groups that generate plankton imaging data to
support this development by contributing datasets from the currently supported instruments. Our pipeline is easily transferable,
in that other imaging instruments and datasets, either new or unpublished, can be ingested in PSSdb, we hence also invite users
of other imaging devices to contact us (info available at https://pssdb.net/) to discuss options.

## 6   Code availability

The Pelagic Size Structure database workflow has been implemented in Python and is freely available at https://github.
com/jessluo/PSSdb.

## 7   Data availability

The first release datasets for the Pelagic Size Structure database project are available at https://doi.org/10.5281/zenodo.10150020
(Dugenne et al., 2023). Further information about the PSSdb project can be found at https://pssdb.net/.





**Figure A1.** Distribution of the sampling depth ranges of accessible (all bars) and ingested (black bars) IFCB (a), scanners (b), and UVP (c) datasets. Note that depth limits were rounded to a 10, 50 and 100 m resolution to reduce the number of ranges reported.

■Biovolume-area  ■Biovolume-distance-map  ■Biovolume-ellipsoid

**Figure A2.** Normalized Biovolume Size spectra (a,b,c) and associated linear regression parameters (d,e,f) calculated from three methods: area-based biovolume (this study), distance maps-based biovolume (Moberg and Sosik, 2012), which are part of the processing pipeline of IFCB images, and ellipsoidal biovolume, which are more commonly used for processing ZooScan and UVP datasets. Dots represent individual samples (defined by temporal and spatial bins), solid lines in panels a, b & c represent the median spectrum for the size classes that were present in at least 50% of the samples (to avoid misalignments due to different sampling efforts). Violin plots in panels d,e & f represent data density on the Y-axis, and horizontal lines represent the median. The data included in this analysis is restricted to particles that have length estimates for both the major and minor axis, resulting in only large particles uploaded on EcoTaxa for UVP datasets.

*Author contributions.* MD, MC-U, JYL, RK, TDO'B, J-OI, FL, LS, CS contributed to the conception and primary efforts on data compilation, quality control and computation leading to the releases and publication of the Pelagic Size Structure database. RK, J-OI, FL, LS, CA, AC, LG, CG, HH, L K-B, RMK, AMcD, MN, MP, J-BR and HMS led the data acquisition. MB, NB, SB, FC, ETC, PD, CD, LD, AE, AF, NG, P-:G, KH, JAH, LJ, KMK, ML, CM, ZM, BN, TP, EP, ER, CR, GS, JT, CT, MV contributed to the data collection, acquisition, analysis or curation. All authors contributed and approve of the manuscript.




*Competing interests.* The authors declare no competing interest.


*Acknowledgements.* This work was mainly funded by NOAA (Award NA21OAR4310254 to JL, RK, LS, FL, J-O I, T o'B and CS) for the project "Developing PSSdb: a Pelagic Size Structure database to support biogeochemical modelling". MD, RK, and LS received further support from the European Union project TRIATLAS (European Union Horizon 2020 program, grant agreement 817578). RK additionally acknowledges support via a Make Our Planet Great Again grant from the French National Research Agency (ANR) within the Programme
d'Investissements d'Avenir #ANR-19-MPGA-0012 and from the Heisenberg Programme of the German Science Foundation #KI 1387/5-1.





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
