# Peer review of "First release of the Pelagic Size Structure database: Global datasets of marine size spectra obtained from plankton imaging devices"

_Earth System Science Data, 2023_

## Author Comment (AC1)

Response to reviews of Dugenne, Corrales-Ugalde et al. "First release of the Pelagic Size Structure database: Global datasets of marine size spectra obtained from plankton imaging devices" submitted to Earth System Science Data (manuscript essd-2023-479)

**Reviewers' comments are pasted in grey, response are written in black**

General response to RC1 and RC2: We thank the reviewers for the insightful comments regarding the quality control of the Size spectra data product, referred in the PSSdb nomenclature as 1b. To alleviate the reviewers concerns, we have implemented a quality control function that flags size spectra parameters where (1) the Normalized biovolume slope exceeds by three times the standard deviation of the global slope mean, (2) The size spectra parameters were calculated with 4 or less size classes and (3) the determination coefficient is below 80%. We hope that these checks can help users in deciding whether to use or discard data points with these flags.

Response to RC1:

Dugenne, Corrales-Ugalde and others provide in this manuscript and the associated dataset a novel and significant contribution to the field. This represents an impressive undertaking and it has been very well executed with care taken to make the data product easy to use and well-documented.

I wholeheartedly support the publication of this manuscript with edits to address my comments below, and some copyediting to catch grammatical errors and make the text clearer and more concise.

We thank Reviewer 1 for their support and provide details on how we addressed their comments below.

Major comments:

Manuscript

The Introduction includes a large amount of relevant information and background on the motivation for developing PSSdb. I would suggest, however, that it would be helpful to the reader/data user to shorten the Introduction by focusing directly on the technologies and previous work directly relevant to PSSdb (e.g. imaging tools, observing systems/platforms, data analysis pipelines), and to check for redundancy/repetition with the sections in the Methods giving background on the different instruments.

We considered the suggestion to reduce the background information in the introduction, but felt that it was essential to start by explaining why the size of pelagic organisms is such an important trait before mentioning the development of known imaging tools, observing programs, and data pipelines. Indeed, ESSD targets many disciplines, with a general audience, so we felt that the manuscript needed to start with a general introduction.

We have also carefully checked for repetition or redundancy in the Methods section 2.1, dealing with instrument-specific acquisition strategy and pre-processing steps. Since such considerations were important, but not necessarily fundamental to understanding the methodology behind the database generation, we decided to move these sections, after cleaning potential repetitions, to supplementary material A0. Instead, we wrote a general paragraph following the Reviewer's next suggestion (see following paragraph).

As written, there is a fair amount of repetition in the Methods section on how images are captured by each instrument and then processed. I would suggest that the authors aim to condense this by providing a general overview of the process of image capture and segmentation/ROI extraction before discussing each instrument individually to make the text clearer and more concise. Details of the process specific to each instrument can then be highlighted in their respective sections.

We have included the suggested paragraph, which now reads in lines 124-151:
"[General intro of the Material and Methods section] In the following sections, we first highlight the key aspects of data acquisition and preprocessing by the three imaging instruments considered in PSSdb (section 2.1). Then

we provide details on the current pipeline for PSSdb ingestion that enables the computation of instrument-specific size spectra currently available at https://doi.org/10.5281/zenodo.10809693 (section 2.2).

[Start of section 2.1: Acquisition and preprocessing steps of imaging datasets] Datasets from several plankton imaging systems were included in PSSdb: the IFCB (Olson and Sosik, 2007), the UVP (UVP5, Picheral et al. (2010), and UVP6, Picheral et al. (2022)) and benchtop Scanner systems such as the Zooscan (Gorsky et al., 2010) and other generic scanners (Gislason and Silva, 2009). In addition to the detailed description provided in their associated publications, further considerations of these instruments deployments and operational specifications relevant to the generation of the database are provided in Supplementary material A0. Here, we provide a brief overview of the main principles guiding image acquisition and pre-processing steps, leading to the ingestion of mentioned imaging datasets in PSSdb.

All instruments were designed to image plankton or particles *in situ* or in the laboratory based on user-defined thresholds (e.g. minimum size for all instruments, laser-induced fluorescence or scattering for IFCB, or pixel intensity for UVP and scanners). Prior to their use, instruments are generally calibrated to ensure that particles detected can be effectively sized (by measuring the pixel size) and counted in a quantitative volume (e.g. calibrated syringe for IFCB, dimensions of the illuminated frame for UVP, and flow-meters mounted on nets for scanners). Particles that pass these thresholds are then segmented (i.e. the process of identifying target particles from background pixels) in near-real time to produce cropped thumbnails of Regions of Interest (ROIs). These thumbnails are automatically saved on the computer piloting the instrument for further processing. Notably, common processing steps across all imaging instruments include the automated identification of pixels enclosing these ROIs (with instrument-specific algorithms) to compute morphometric features, including area or ellipsoidal axis as well as pixel intensity descriptors. These can be used to train machine learning algorithms which predict taxonomic annotations of the entire set of ROIs, although new classifiers now directly use the thumbnails and extract their own "features". Thumbnails, morphometric features, and potential taxonomic annotations are then all uploaded to online platforms, such as EcoTaxa/Ecopart for scanners and UVPs or dashboards for IFCBs, that are not long-term storage repositories *per se* but help to visualize and check incoming datasets or curate the classifier predictions by taxonomic experts (in the case of EcoTaxa). Importantly, all datasets are typically uploaded with sufficient metadata, comprising the GPS coordinates, sampling time, camera pixel size and calibrated volume, to support their ingestion in large data aggregation projects like PSSdb. We only selected datasets with taxonomic annotations for the generation of PSSdb, to ensure that bulk size spectra did not include methodological artefacts like bubbles or calibration beads, and for further work on taxon-specific data products."

As mentioned in the previous answer, we have decided to move the instrument-specific descriptions to appendix A0, since we consider that there are key distinctions between the instruments that informed PSSdb development, and some readers might find these details useful.

The graphical representation of the PSSdb data pipeline (Figure 1) is nicely done and a helpful summary for users of the data. However, it doesn't show how taxonomic annotations are used to filter the data prior to ingestion. This is an important part of the pipeline and I think an essential part of the process so I would like to see it indicated in panel (a) of Figure 1 somehow.

We thank Reviewer 1 for this suggestion. We have now updated Figure 1 and mention that only datasets with taxonomic annotations were downloaded "to ensure that bulk datasets do not include artefacts, and to generate taxa-specific products" (Figure 1 caption).

Since you refer in the Methods section to particular scripts within the PSSdb pipeline, please include a direct link to the PSSdb GitHub repository in Section 2.2. I see that there is a link to the repo in the Code Availability statement and links to particular scripts given in Section 2.2.2, but it would be helpful for the reader/data user to have a link to the full repository given concurrently with the overview of the whole pipeline.

We agree with Reviewer 1 and have now included a reference to PSSdb GitHub repository in the general introduction of section 2.2 (lines 159-160): "All steps are associated with a numbered script coded in Python, fully available at https://github.com/jessluo/PSSdb."

Section 2.2.3: Could you elaborate on what exactly you mean when you say that the standardizer spreadsheets need to be "filled manually". It would help here to have a specific example of a manual operation that is executed as part of the pipeline. I see that you state that information needs to be "manually filled to map the native headers and units", but I don't feel that this is sufficiently descriptive for a reader/data user to understand the process and the extent of manual data manipulation. For example, when dealing with units, are you simply applying a conversion factor to a column of data, typing numbers into blank cells, or copy/pasting from another spreadsheet?

Thank you for pointing this out. We now include more details regarding this manual step, by precising that : "After listing and exporting all datasets from EcoTaxa, EcoPart, or IFCB dashboards, a member of PSSdb thus enters the name and corresponding unit found in the native export files to each variable needed in future steps of the pipeline so that they can be mapped and converted to the standards defined in the products documentation. This mapping and conversion is then done automatically using the script developed for the standardization (https://github.com/jessluo/PSSdb/blob/main/scripts/2_standardize_projects.py)." (lines 220-224). For units conversion, we mention that: "Native units, defined in the standardizer spreadsheets, are converted to standard units using the python package Pint (https://pypi.org/project/Pint/), designed to define, operate and manipulate physical quantities, based on units from the International System or defined in a custom text file." (l. 238-240). For example, one just have to fill in the native unit - "milliliter" for IFCB volume sampled column initially named "vol_analyzed"- to automatically convert it into a variable in liter (PSSdb products are expressed in unit biovolume per liter per unit biovolume).

I was curious about the lack of flags related to the quality of derived products (NBSS/PSD slope, intercept, etc…). I have accessed and worked with the PSSdb data and in some (few) instances have found IFCB PSD slopes approaching -9, which seems pretty steep. It may be that there isn't enough data available to come up with QARTOD style data flags, but it would be useful to know how the PSSdb team might try to address this in future.

We agree with the reviewer that some IFCB data might be questionable. After considering all reviewers' comments, we decided to include more quality control checks in our pipeline. The first one uses the flag annotations in the IFCB dashboards (referred to as "tag") that labels the data as having bad focus, data from a culture, a calibration run, among other labels to discard faulty data. This check is implemented during the downloading step, to save storage space. In addition, we have included a QC process when generating PSSdb final products (see general comment). The new products have been uploaded on Zenodo (https://doi.org/10.5281/zenodo.10809693), and the revised manuscript now describes this additional QC on lines 347-351 :"All products (1a: size spectra, 1b: regression parameters) generated are subject to an additional QC, to provide a flag (0 if a spatio-temporal bin passed the QC, 1 otherwise) that can help data users filter out questionable data. The current QC is based on 3 criteria, whereby a positive flag is assigned to (1) slopes values exceeding the mean ±3 standard deviations of each instrument-specific product, (2) spectrum that only record four or less non-empty size classes, and (3) log-linear fit whose regression fit $R^2 \leq 0.8$."

I wonder if the authors could comment on why they opted to aggregate the UVP data over the profile rather than separate it by depth (as done by Kiko et al., 2022)? I can see advantages (allows for comparison and/or integration with ZooScan data product) and disadvantages (loses valuable information on size structure patterns with depth) to either approach, but I think that it would be helpful to unpack this a little more in the manuscript, particularly for readers/data users who might want to use both the PSSdb and the Kiko UVP datasets for their research.

Our primary goal with this aggregation strategy was to produce robust size spectra estimates, based on a sufficient number of particles determined statistically. Even though this did not constitute a strong limitation for the bulk datasets as presented in this manuscript, future work involving taxa-specific datasets would not have passed our QC without aggregating certain groups of particles between 0-200 m. We thus integrated UVP profiles with depth to be consistent amongst products. Second, in future releases we are hoping to generate merged products across multiple instruments, and the discrepancy in the vertical resolution of samples collected by the three instruments encouraged us to take this approach, as mentioned by Reviewer 1 and written in the discussion on lines 592-603:"In some cases, merging imaging datasets integrated over specific depth layers (e.g. net-collected datasets) with profiling or towed datasets is facilitated by simply integrating observations using the lowest sampling resolution (Soviadan et al. 2023). [...] We hope to build upon this first data release and recent work from Soviadan et al. (2023) to provide merged data products that will effectively span the five orders of magnitude that can be captured by commercially available plankton imagers (Lombard et al. 2019). Further, we are planning on releasing taxonomically resolved PSSdb products [...]".

Data Quality:

I have downloaded, accessed and inspected data from both files associated with all three instruments. The .csv data files are well organised, and the documentation provided is clear and provides the necessary information to understand the content of the data files. I had no issues loading the files into python for analysis and visualisation.

Thank you for testing this.

Minor comments:

L11 (and throughout the document): should be 1°x1°. Corrections have been made throughout the document.

L12 and L393: "Our instrument-specific datasets span all major ocean basins" I agree that the coverage of PSSdb is impressive, but this statement somewhat overstates the case. Please mediate your language to provide a statement that better reflects the actual data coverage. Suggest something like: "PSSdb includes data collected in the major ocean basins…". We have corrected these sentences to: "These datasets span most major ocean basins, although all basins are undersampled in the southern hemisphere".

L12-13: IFCBs have been deployed in regions other than the northern latitudes, that data simply isn't included in your data product. We now state on lines 12-14: "Our instrument-specific datasets span most major ocean basins, except for the IFCB datasets we have ingested that were exclusively collected in northern latitudes, and cover decadal time periods [...]"

L43: should be "reviews". Corrected

L51 – 61: although it is probably considered common knowledge (for biological oceanographers), since it is likely that this dataset will be used by a wider range of researchers, please provide the size range associated with each group you mention (nano-, micro-, meso-, macro-plankton and micronekton). We now provide the definition of plankton size classes defined by Sieburth early in the introduction, on lines 28-32: "These studies call for a global assessment of the plankton size continuum, rather than the discrete size categories defined by Sieburth et al. (1978) (i.e. picoplankton: 0.2-2 μm, nanoplankton: 2-20 μm, microplankton: 20-200 μm, mesoplankton: 200-20,000 μm, nekton: 2,000-20,000,000 μm), to study ecosystem functioning or to model ecosystem services under current and future anthropogenic pressures (Lombard et al. 2019, Ljungstrom et al. 2020, Atkinson et al. 2021). "

L72: please also include a reference to Miloslavich et al. (2018) here as it is a key reference for Essential Biodiversity Variables (EBVs)

Miloslavich, P., Bax, N.J., Simmons, S.E., Klein, E., Appeltans, W., Aburto-Oropeza, O., Andersen Garcia, M., Batten, S.D., Benedetti-Cecchi, L., Checkley Jr, D.M. and Chiba, S., 2018. Essential ocean variables for global sustained observations of biodiversity and ecosystem changes. Global Change Biology, 24(6), pp.2416-2433. The suggested reference has been included

L73: this should Bio-GO-SHIP and BGC-Argo, not "or" Corrected

L75: this sentence needs tidying up, it's a little unclear what point you are trying to make as you are mixing up instruments (IFCB, ZooScan) and time series (CalOOS, NESLTER, Point B).

We have modified this paragraph to better make our point that plankton imaging systems provide a unique opportunity to generate datasets on the key ocean variables mentioned in lines 72-73, and because of this, they are, and will be part of observing programs that cover large scales of space and time.

L87: please fix the grammar in this sentence, should not be "allowed to" Corrected

L88: If you are going to invoke FAIR data practices, then you need to reference Wilkinson et al (2016) here (rather than L91 below). Corrected

L90: Suggest that you open directly with a sentence telling the reader what PSSdb is, i.e. move up the text from lines 95-98, and then give the context around the product (lines 90-95). The paragraph has been modified

L93: what do you mean by "consistent across the 7-10,000 um size range"? Readers may interpret this to mean that you are presenting data that has consistent coverage across this range, but that isn't the case as not all of the instruments included in PSSdb cover that range, and nor were they all deployed in concert. By consistent, we meant that the spectra aligned to a great extent across the instruments and specific target sizes. We have now modified the sentence to clarify that this is the size range that can be sampled by imaging systems on lines 98-100: "Our project capitalizes on largely untapped size structure observations from plankton imaging devices,

which can image plankton and particles across the 7-10,000 µm size range (Romagnan et al. 2015, Lombard et al. 2019)"

L103: should be "includes" **Corrected**

L107: suggest that you update to "have been adopted to represent the exponential decrease in particle abundance typically observed as size increases" **The sentence has been updated**

L108: "respectively" is not necessary here. **Corrected**

L133: you should qualify this statement to add that "most IFCB sampling efforts included in PSSdb are limited to…" **The sentence has been modified. Note that this and the next three comments are now in the Appendix.**

L152: rather than repeat depth rating for UVP5 and UVP6, state it once at end of sentence for both instruments. **Corrected**

L184: "respectively" is not necessary here as the sentence suggests that all of the scanners of interest have the same resolution. **Corrected**

L184: Suggest saying "These scanners" rather than "Both" for clarity **Corrected**

L208: you have one too many "and"s in this sentence **Corrected**

L221: please define what is meant by "methodological artefacts" since this is an important point that comes up quite a bit with respect to QC. **We have re-written this sentence to emphasize the use of taxonomic annotations for discarding particles that are not useful for size spectra calculations and mention examples of artefacts (bubbles, calibration beads, etc.)**

L242: give the name of this script as you have with other scripts above. **The link to the script is now included**

L281: please provide a reference or URL to the documentation for the Pint package **The link is now included**

L304: Suggest that you include a reference to Haentjens et al (2022) here as that paper has a very nice section in the methods and supplementary materials (S4) describing the process of determining the count uncertainty. **The reference is now included**

L354: suggest moving the Kiko citation in the sentence, as written it implies that the Kiko study is related to satellite algorithms, which it isn't. **Corrected**

L417-422: This is repetition of introductory material, suggest removing and focusing only on results here. **We consider that it is appropriate to include a sentence that highlights which plankton size classes following Sieburth et al. (1978) are represented in the data products. However, we have moved the sentence regarding better measurements of non living particles and gelatinous plankton in UVP to the discussion (lines 505-507).**

L531-533: Do you mean that by restricting the settings to a narrower size/fluorescence threshold that the IFCB data would be more reliably from phytoplankton cells? Would help to clarify this point a bit here, particularly since the preceding sentence is about spatial coverage rather than taxonomic annotation. **Lines 488-490 have been modified to better communicate that narrower PMT settings will decrease the number of particles imaged and will result in more manageable datasets. In addition, the paragraph in lines 484-496 has been modified to better highlight some aspects that can increase the inclusion of IFCB data in PSSdb.**

L544" should be "divergences from…" **Corrected**

L564; Do you mean "Such shifts could be detected…". This sentence is a bit of a leap from the previous one. It might be clearer to change the order of the clauses and make the point that the PSSdb data could provide a baseline from which deviations (such as HAB events) could be detected with ongoing monitoring (I think that this is the idea you were trying to convey here). **We have modified the sentences in lines 529-536 to clarify that time series datasets of size spectra parameters are useful to understand plankton community dynamics, especially when comparing anomalies to the complete time series.**

Figure A1: the depth ranges on the x-axis are very hard to see, suggest rotating the axis tick labels to 45° to make them easier to read. **We have tried rotating the labels to 45°, but the labels were even less visible, so we kept the same layout.**

Figure A2: please include a legend indicating what each colour represents in the violin plot, and also include this information in the figure legend. **The suggested legend and the textual description in the figure legend have been included**

**Response to RC2:**

Summary of the manuscript and significance:

Dugenne et al. have achieved here what I would call the "holy grail" of size-spectra research: Gathering the particle raw data captured from different imaging systems and around the globe, and building a pipeline to standardize and process all this information to construct size-spectra under the same methodology. Size-spectra produced this way are intercomparable, especially their intercepts that are heavily influenced by the choice of units and bin size. The methodological pipeline the authors developed, and presented here, is meticulous and flexible enough that will allow the integration of other datasets and from other imaging systems than the ones used in this study. Therefore, this manuscript and its corresponding database is unique in this regard, compared to other global-scale size-spectra datasets (like in Atkinson et al. 2024, https://doi.org/10.1038/s41467-023-44406-5 ). I expect that the database provided here by Dugenne et al. is going to be invaluable for global-scale research in the field.

**Thank you so much, we greatly appreciate your encouraging words.**

The manuscript is generally very well written, and the supporting documentation found in the dataset and github repositories is comprehensive and helpful.

I strongly support the publication of this manuscript after addressing the points raised below. I will avoid repeating points raised by the first Referee, unless I believe they should be restressed or have something to add to them. After the major and minor comments, I will provide some suggestions for the future development of the database that the authors should not feel obliged to address.

**Thank you for taking into consideration the other review and expanding on their comments, as well as providing useful suggestions.**

Major comments:

Point 1 (critical): I've encountered some potential errors in the size distribution datasets. The five largest biovolume size classes in the Scanner and UVP datasets ($6.14 \times 10^{12}$ to $4.91 \times 10^{13}$) have been constructed using a different size step than the rest. I.e. $V_{n+1}/V_n \neq 2$ or $\log_2(V_{n+1}) - \log_2(V_n) \neq 1$ (where V = biovolume size class) for said classes. Is this possibly due to something going wrong when merging size-fractioned data? Additionally, there are also some issues in the lower or upper size ends of some spectra. There are some cases when the largest class/classes should have been dropped (according to what the authors describe in lines 375 – 377) but they didn't (maybe this is somehow related to the above-mentioned issue?) (see [i]). Also, according to authors, the size bin of the maximum biovolume/abundance was chosen as the lower size limit (lines 379 – 380), but there are cases where this is not true (see [ii]). These are all critical issues that can potentially heavily influence the spectrum parameters and should be addressed before publication. Regrettably, this means that the corresponding graphs and analyses in the results should also be checked and corrected if necessary (but, see also Point 2).

**We thank reviewer 2 for checking carefully the products and providing detailed examples of faulty datasets below. We have identified the cause for the discrepancies: the thresholding function described on lines 325-337, and have now corrected it, checking the expected datasets provided in [i]. As a consequence, we have now updated the products on Zenodo and all the product links, statistics and figures provided in the manuscript. Although we had to re-generate the products, we note that none of the updated statistics and trends presented or discussed in the manuscript differed significantly from the previous estimates (mostly the 2nd decimals) .**

The low biovolume/abundance in the smallest size classes observed in [ii] results from aggregating and averaging norrmalized biovolume/abundances in product 1a. As mentioned in lines 284-292, size spectra parameter calculations were performed on weekly, 0.5°x0.5° grid cells, and afterwards, both the size spectra themselves and their parameters were averaged in monthly, 1°x1° grid cells. Thus, it is likely that some size classes were not present in all sub-bin estimates that were averaged in the monthly 1-degree cells, resulting in the pattern observed by the reviewer. We have included a clarification regarding this pattern in lines 337-339: "It is important to clarify that this thresholding is applied to the weekly, 0.5°x0.5° bins, so it is possible that 1a products present low normalized/abundance values at the lower end if the smallest size class is present in only some sub-bins"

[i] (dataset_ocean_year_month_latitude_longitude) Scanner_Mediterranean Region_2017_9_43.5_7.5, Scanner_South Atlantic Ocean_2010_10_-12.5_-24.5, Scanner_South Atlantic Ocean_2019_10_-0.5_-22.5, Scanner_South Atlantic Ocean_2019_10_-0.5_-23.5, Scanner_South Atlantic Ocean_2019_10_-0.5_-25.5, Scanner_South Pacific Ocean_2011_5_-2.5_-84.5, UVP_South Pacific Ocean_2019_-12.5_-77.5 and maybe Scanner_Mediterranean Region_2015_4_43.5_7.5, IFCB_North Atlantic Ocean_2016_7_40.5_-72.5.

[ii] Scanner_Indian Ocean_2010_4_-0.5_73.5, Scanner_Indian Ocean_2013_7_-25.5_47.5, Scanner_Indian Ocean_2019_3_-34.5_23.5. Scanner_Indian Ocean_2019_3_-34.5_25.5, Scanner_Mediterranean Region_1997_11_43.5_7.5, Scanner_Mediterranean Region_1997_6_43.5_7.5, Scanner_Mediterranean Region_2000_5_43.5_7.5, Scanner_Mediterranean Region_2014_11_43.5_7.5, Scanner_North Atlantic Ocean_2006_5_44.5_-1.5, Scanner_North Atlantic Ocean_2006_5_44.5_-2.5, Scanner_North Atlantic Ocean_2009_5_44.5_-2.5, Scanner_North Atlantic Ocean_2009_5_47.5_-4.5, Scanner_North Atlantic Ocean_2010_4_43.5_-1.5, Scanner_North Atlantic Ocean_2010_5_47.5_-2.5, Scanner_North Atlantic Ocean_2010_5_47.5_-4.5, Scanner_North Atlantic Ocean_2011_4_44.5_-1.5, Scanner_North Atlantic Ocean_2011_5_46.5_-3.5, Scanner_North Atlantic Ocean_2013_5_45.5_-3.5, Scanner_North Atlantic Ocean_2014_4_44.5_-2.5, Scanner_North Atlantic Ocean_2014_5_45.5_-2.5, Scanner_North Atlantic Ocean_2014_5_46.5_-3.5, Scanner_North Atlantic Ocean_2014_5_47.5_-4.5, Scanner_North Atlantic Ocean_2015_5_46.5_-4.5, Scanner_North Atlantic Ocean_2015_5_47.5_-3.5, Scanner_North Atlantic Ocean_2016_5_44.5_-1.5, and maybe Scanner_Mediterranean Region_2019_9_43.5_7.5, Scanner_North Atlantic Ocean_2012_5_46.5_-2.5, Scanner_North Atlantic Ocean_2012_5_46.5_-4.5, Scanner_North Atlantic Ocean_2012_5_47.5_-4.5, Scanner_North Atlantic Ocean_2013_5_47.5_-4.5, Scanner_South Atlantic Ocean_2014_5_-0.5_-22.5. Some IFCB spectra also have this issue.

Point 2 (major): As the first Referee pointed out, there is a lack of flags regarding the quality of the size-spectra. When parsing through the data, I understood that this is most probably a very tricky task to untangle. The easiest quality flag that can be assigned is removing the spectra generated from a very low number of size classes. For example, scanner spectra with less than five size classes should be dropped and excluded from the final dataset (e.g. Scanner_Mediterranean Region_1996_9_43.5_7.5, Scanner_Mediterranean Region_2000_2_43.5_7.5 and others). Accordingly, an appropriate threshold should be used for the UVP and the IFCB datasets. Regarding the fact that some spectral slopes in the datasets are very steep or very shallow, I trust that the methodology the authors followed should have otherwise caught any erroneous entries. I can imagine that flagging some spectra as "suspect" may be useful for users (e.g. if the slope is very steep/shallow and R2 is low, flag the entry as "suspect"), but it might lead them to simply filter out these entries, potentially loosing information that reflects the true state of the system at the time.

We thank reviewer 2 for stressing this again. We realized that this was a crucial point that could be really valuable for data users. Since we had to update the products, with the corrected thresholding function, we decided to already implement the suggested QC in the updated products. From now on, the products will include 3 flags (0 if a spatio-temporal bin passed the QC, 1 otherwise) that can help data users filter out questionable data if they want while still keeping "suspicious" data, as per your suggestion. "The current QC is based on 3 criteria, whereby a positive flag is assigned to (1) slopes values exceeding the mean ±3 standard deviations of each instrument-specific product, (2) spectrum that only record four or less non-empty size classes, and (3) log-linear fit whose regression fit $R^2 \leq 0.8$." l.394-398

Minor comments:

Point 3 (minor): Line 353 (but also in the database repository and supporting documentation): San Martin et al. (2006) used normalized biomass size-spectra in their publication, not biovolume. Perhaps another citation is more fitting here. We have replaced that reference with Zhang et al (2019) and Grandrémy et al (2023).

Point 4 (minor): In the beginning of the section 2.2.1 (Selection of imaging data streams), please mention again that the generated particle size-spectra are bulk, i.e. they include both "living" and "non-living" particles. In the next paragraph (or if you choose to mention "living" and "non-living" at the beginning), provide a couple of examples of what are considered to be representative particles for each category. For example, "[...] living (i.e. planktonic and micronektonic organisms, eggs, etc.) and non-living (e.g. marine snow, aggregates, fecal pellets, etc.) particles […]". Similarly, in line 221, please elaborate on what are considered methodological artefacts, as the first Referee suggested. A comprehensive list is included in the authors' github repository (plankton_annotated_taxonomy.xlsx), but nevertheless, it should be also clear in the text. We have provided examples of living and non living particles, as well as examples of particles that were excluded based on annotations in lines 170-173.

Point 5 (minor): Lines 327-328: I suggest rephrasing the sentence "Since individual weeks could occur in two separate months, we assigned a unique month to each week by selecting the month that counted most samples" to something a bit clearer. E.g. "Since unavoidably a particular week might be shared between two months, we assigned that week to the month that counted most samples". We have rephrased the sentence in lines 284-285 following the reviewer's suggestion.

Point 6 (minor): When discussing slopes, I would avoid using descriptions such as "decreasing/increasing" or "lower/higher" that can be interpreted both ways, and instead use "steep/shallow" wherever possible. There is no confusion in the text currently, but mainly because the authors use both descriptions in tandem (e.g. in line 464: "[…], whose NBSS slopes decreased (i.e. steepened)[…]"). In most cases I'd simply use only the non-arbitrary "steep/shallow".

We agree and have deleted mentions of 'decrease/increase' of the spectral slopes throughout the text to only refer to "flatter/steeper" spectra like so: "Indeed, while the majority of the slopes were around -1 L$^{-1}$ µm$^{-3}$, the scanner slopes showed no clear variation with space. Meanwhile, the UVP slopes tended to show steeper size spectra within oligotrophic gyres and flatter size spectra in the northernmost latitudes or by the coasts (Fig. 4c). This pattern was inverted with regards to the intercepts, as the abundance of 1 µm$^3$ particles was lower in the Arctic and increased near shore (Fig. 4f). Likewise, the IFCB NBSS slopes were indicative of flatter size spectra, with lower intercepts, in the northernmost latitudes and along the Eastern coast of the United States, compared to the Western coast (Fig. 4a,d)." lines 398-403.

Point 7 (minor): Lines 474 – 491: The decoupling between slopes and intercepts is by itself very interesting, but what is missing here is a clearer mention that these parameters are known to be (in general) strongly inversely correlated (shown in Gómez-Canchong et al. 2013, https://doi.org/10.3989/scimar.03708.22A , and mentioned in Sprules and Barth 2016 review, https://doi.org/10.1139/cjfas-2015-0115 ). As of now, this is left at an "often observed" in line 474 and an "As expected" in line 479. Daan et al. (2005, https://doi.org/10.1016/j.icesjms.2004.08.020 ) used the mid-point "height" of the spectrum (or "elevation" in Atkinson et al. 2021) to avoid this issue (including the mid-point height is one of my suggestions for future development of the database).

We agree and have replaced "often" by "typically" to stress this further. Lines 432-435 now reads: "Given the opposite spatial (Fig. 4,5) and temporal (Fig.6) patterns typically observed between the size spectra slopes and intercepts (Sprules and Barth, 2016), across instruments and oceanic regions, we used the yearly time series correlation of these 2 parameters in any given grid cell within the same oceanic region as a way to detect potential decoupling, lag, or feedback between the two."

Point 8 (minor): There is also a recent publication by Atkinson et al. (2024, https://doi.org/10.1038/s41467-023-44406-5 ) where the authors presented a global-scale normalized biomass size-spectrum slopes compilation, covering a range of organisms of at least 7 orders of magnitude and including marine and lake ecosystems. The authors showed that steeper size spectra with decreasing phytoplankton biomass indicate strong trophic

amplification that may materialize into significant fish biomass declines under future warming scenarios. I believe the inclusion of this article will be an interesting addition to the discussion.

We now mention the compilation of Atkinson et al. in the discussion on lines 462-464: "More recently, Atkinson et al. (2024) compiled estimates of spectral slopes measured in 41 sites, mostly located in the Atlantic ocean and a number of lakes, displaying important characteristics relevant to study the impact of climate change."

Point 9 (minor): In the database repository and the supporting documentation the authors mention that the Normalized Biomass (instead of Biovolume) Size Spectra are included (dot 2). Please correct this accordingly.

The computation of normalized biomass size spectra is restricted to taxa-specific products, which we have now released (https://zenodo.org/records/10810191), since they require applying taxa-specific biovolume-to-biomass conversion factors. We have revised the last paragraph of the discussion to mention these new products: lines 606-609: "These products, now available at https://zenodo.org/records/10810191 and described in Dugenne et al. (2024),  incorporate different levels of taxonomic resolution, allowing a global assessment of group-specific size structure and derived biomass based on published relationships linking biovolume to carbon content  (Menden-Deuer and Lessard 2000, Lehette 2009, McConville 2017)."

Suggestions:

As I mentioned in the beginning, these are some suggestions I have for the future development of the database that the authors are not obliged to address. Several future endeavors I would propose are covered by the authors in the discussion, namely: a) The inclusion of normalized biomass size-spectra in the database (the authors made good progress on this, as they have already compiled plenty of size-to-mass equations in the github repository), b) the inclusion of merged data products and c) the release of taxonomically resolved PSSdb products (what I'd like to see here is simply separate "living" and "non-living" particle size-spectra).

1) Probably another thing to consider is including the spectrum mid-point height in the products (as I mentioned in Point 7), a parameter that is generally uncorrelated with the spectrum slope, which I believe might be useful for researchers in the field.

We thank the reviewer for this suggestion, and agree that this could be useful to implement in the future. However, we note that one can already test and generate such estimates from our products, since we do provide the full spectra in 1a datasets.

2) There is a discussion in the field regarding the spectrum slope calculations, initiated (to the best of my knowledge) by White et al. 2008 ( https://doi.org/10.1890/07-1288.1 ) and expanded by Edwards et al. (2017, https://doi.org/10.1111/2041-210X.12641 ), where the use of maximum likelihood estimation is advised. In Edwards et al. 2020, ( https://doi.org/10.3354/meps13230 ) a methodology for estimating slopes through MLE using binned data is also described. However, Barth et al. (2019, https://doi.org/10.1139/cjfas-2018-0371 ) found that the MLE can be biased when non-linear structures (i.e. domes and/or valleys) are present in the spectra. Since the authors have access to the raw unbinned data, it would be nice if they could explore this in the future.

We are aware of the work and package form Edwards et al., and as mentioned for the previous suggestion, gladly invite data users to test different algorithms using products 1a if they are interested in doing so.

3) I believe that translating the python scripts into R with the goal of finally building a comprehensive R package will significantly boost visibility and engagement from the community.

Our focus to expand the database in the future will likely center towards ingesting new instruments and generating merged products, but we will keep this suggestion in mind as some PSSdb team members are also strong R supporters. However, the given funding situation for PSSdb currently does not allow the translation of our code to R.

Cited References

[revised manuscript text omitted]

---

## Referee Report (RR1)

Review of manuscript **essd-2023-479** from **Dugenne *et al.*** with the title "**First release of the Pelagic Size Structure database: Global datasets of marine size spectra obtained from plankton imaging devices**"

**2nd Round of the review.**

*General*: I commend the authors for their responses and the changes made to the manuscript. I have no further comments on the revised manuscript.

However, I need to restress a point missed from my first comment.

Major comments:

*previous comment:* Point 1 (critical): I've encountered some potential errors in the size distribution datasets. The five largest biovolume size classes in the Scanner and UVP datasets ($6.14 \times 10^{12}$ to $4.91 \times 10^{13}$) have been constructed using a different size step than the rest. I.e. $V_{n+1}/V_n \neq 2$ or $\log_2(V_{n+1}) - \log_2(V_n) \neq 1$ (where $V$ = biovolume size class) for said classes. Is this possibly due to something going wrong when merging size-fractioned data? Additionally, there are also some issues in the lower or upper size ends of some spectra. There are some cases when the largest class/classes should have been dropped (according to what the authors describe in lines 375 – 377) but they didn't (maybe this is somehow related to the above-mentioned issue?) (see [i]). Also, according to authors, the size bin of the maximum biovolume/abundance was chosen as the lower size limit (lines 379 – 380), but there are cases where this is not true (see [ii]). These are all critical issues that can potentially heavily influence the spectrum parameters and should be addressed before publication. Regrettably, this means that the corresponding graphs and analyses in the results should also be checked and corrected if necessary (but, see also Point 2).

*authors' response:* We thank reviewer 2 for checking carefully the products and providing detailed examples of faulty datasets below. We have identified the cause for the discrepancies: the thresholding function described on lines 322-337, and have now corrected it, checking the expected datasets provided in [i]. As a consequence, we have now updated the products on Zenodo and all the product links, statistics and figures provided in the manuscript. Although we had to re-generate the products, we note that none of the updated statistics and trends presented or discussed in the manuscript differed significantly from the previous estimates (mostly the 2nd decimals). The low biovolume/abundance in the smallest size classes observed in [ii] results from aggregating and averaging normalized biovolume/abundances in product 1a. As mentioned in lines 284-292, size spectra parameter calculations were performed on weekly, 0.5°x0.5° grid cells, and afterwards, both the size spectra themselves and their parameters were averaged in monthly, 1°x1° grid cells. Thus, it is likely that some size classes were not present in all sub-bin estimates that were averaged in the monthly 1-degree cells, resulting in the pattern observed by the reviewer. We have included a clarification regarding this pattern in lines 334-336: "It is important to clarify that this

thresholding is applied to the weekly, 0.5°x0.5° bins, so it is possible that 1a products present low normalized/abundance values at the lower end if the smallest size class is present in only some sub-bins".

*response to authors:* The authors missed one of the three issues I've raised in my first point. Perhaps this was due to bad wording of my original comment, because, otherwise, they thoroughly and carefully addressed this point. The issue not addressed is the fact that some of the larger size classes in the UVP and Scanner datasets are not consistent with the rest of the size classes, i.e. $V_{n+1}/V_n \neq 2$. To demonstrate this, let's look at the biovolume size class ratio ($V_{n+1}/V_n$) of the UVP and Scanner datasets:

| biovolume_size_class | BV/ratio ($V_{n+1}/V_n$) |
|---|---|
| ... | ... |
| 1.08e+14 | 2.00 |
| 2.16e+14 | 2.00 |
| 4.30e+14 | 1.99 |
| 8.61e+14 | 2.00 |
| 1.73e+15 | 2.01 |
| 3.45e+15 | 1.99 |
| 6.14e+15 | 1.78 |
| 1.03e+16 | 1.68 |
| 1.74e+16 | 1.68 |
| 2.92e+16 | 1.68 |

It is immediately apparent that there is something wrong with the larger size classes. In any case, I've pinpointed the source of the problem:

In the github repository (https://github.com/jessluo/PSSdb): When the NBSSs are computed (*4_compute_NBSS.py*), the *size_binning_func()* is called (included in *funcs_NBS.py*) which then calls the *ecopart_size_bins.tsv* (PSSdb/ancillary) through the *configuration_masterfile.yaml*. The *ecopart_size_bins.tsv* contains the ESD and biovolume size bins and is indeed the source of this error. In both ESD and biovolume, the ratio $n+1/n$ (*n*=size bin) should be constant for all *n*. But from line 45 and after, the $n+1/n$ ratio changes (from ~1.26 to ~1.19 for ESD, and from ~2 to ~1.68 for biovolume). Correcting this file should fix this issue. The dataset should also be updated accordingly.

---

## Author Response (AR2)

Review of manuscript **essd-2023-479** from **Dugenne *et al.*** with the title "**First release of the Pelagic Size Structure database: Global datasets of marine size spectra obtained from plankton imaging devices**"

We thank the reviewer for bringing the issue described below back to our attention. We have corrected the *ecopart_size_bins.tsv* file such that the ratio $n+1/n$ is kept constant from the smallest to the largest size class (https://github.com/jessluo/PSSdb/blob/main/ancillary/ecopart_size_bins.tsv). Given that this file is crucial for the size spectra calculations, we have updated figures 3-7, and figure A2, as well as the summary statistics in Table 2 and throughout the text. Linked to these changes, we have released updated data products found here: https://doi.org/10.5281/zenodo.11050013. We have replaced the old links with this new version in lines 15, 126, 174, 621, and 639, as well as in the legend of Figure 1. Since the taxa-specific products were also updated, a new link is listed in line 600. In addition, we have provided a new link in the abstract (lines 15-16) and data availability statement (lines 639-640) that will always direct the user to the latest version of the data product: https://zenodo.org/doi/10.5281/zenodo.7998799.

**2ⁿᵈ Round of the review.**

*General*: I commend the authors for their responses and the changes made to the manuscript. I have no further comments on the revised manuscript.

However, I need to restress a point missed from my first comment.

*response to authors:* The authors missed one of the three issues I've raised in my first point. Perhaps this was due to bad wording of my original comment, because, otherwise, they thoroughly and carefully addressed this point. The issue not addressed is the fact that some of the larger size classes in the UVP and Scanner datasets are not consistent with the rest of the size classes, i.e. $V_{n+1}/V_n \neq 2$. To demonstrate this, let's look at the biovolume size class ratio ($V_{n+1}/V_n$) of the UVP and Scanner datasets:

| biovolume_size_class | BV/ratio ($V_{n+1}/V_n$) |
|---|---|
| … | … |
| 1.08e+14 | 2.00 |
| 2.16e+14 | 2.00 |
| 4.30e+14 | 1.99 |
| 8.61e+14 | 2.00 |
| 1.73e+15 | 2.01 |
| 3.45e+15 | 1.99 |
| 6.14e+15 | 1.78 |
| 1.03e+16 | 1.68 |
| 1.74e+16 | 1.68 |

2.92e+16 | 1.68

It is immediately apparent that there is something wrong with the larger size classes. In any case, I've pinpointed the source of the problem:

In the github repository (https://github.com/jessluo/PSSdb): When the NBSSs are computed (*4_compute_NBSS.py*), the *size_binning_func()* is called (included in *funcs_NBS.py*) which then calls the *ecopart_size_bins.tsv* (PSSdb/ancillary) through the *configuration_masterfile.yaml*. The *ecopart_size_bins.tsv* contains the ESD and biovolume size bins and is indeed the source of this error. In both ESD and biovolume, the ratio $n+1/n$ ($n$=size bin) should be constant for all $n$. But from line 45 and after, the $n+1/n$ ratio changes (from ~1.26 to ~1.19 for ESD, and from ~2 to ~1.68 for biovolume). Correcting this file should fix this issue. The dataset should also be updated accordingly.